# Learning-Based Link Anomaly Detection in Continuous-Time Dynamic Graphs

**Tim Poštuvan**[*]  *tim.postuvan@epfl.ch*
*EPFL*

**Claas Grohnfeldt**[†]  *claas.grohnfeldt@huawei.com*
*Huawei Technologies*

**Michele Russo**  *michele.russo@huawei.com*
*Huawei Technologies*

**Giulio Lovisotto**[*]  *giuliolovisotto@gmail.com*
*Huawei Technologies*

**Reviewed on OpenReview:** *https://openreview.net/forum?id=8imVCizVEw*

## Abstract

Anomaly detection in continuous-time dynamic graphs is an emerging field yet under-explored in the context of learning algorithms. In this paper, we pioneer structured analyses of link-level anomalies and graph representation learning for identifying categorically anomalous graph links. First, we introduce a fine-grained taxonomy for edge-level anomalies leveraging structural, temporal, and contextual graph properties. Based on these properties, we introduce a method for *generating* and *injecting typed anomalies* into graphs. Next, we introduce a novel method to *generate continuous-time dynamic graphs* featuring consistencies across either or combinations of time, structure, and context. To enable temporal graph learning methods to *detect specific types of anomalous links* rather than the bare existence of a link, we extend the generic link prediction setting by: (1) conditioning link existence on contextual edge attributes; and (2) refining the training regime to accommodate diverse perturbations in the negative edge sampler. Comprehensive benchmarks on synthetic and real-world datasets – featuring synthetic and labeled organic anomalies and employing six state-of-the-art link prediction methods – validate our taxonomy and generation processes for anomalies and benign graphs, as well as our approach to adapting methods for anomaly detection. Our results reveal that different learning methods excel in capturing different aspects of graph normality and detecting different types of anomalies. We conclude with a comprehensive list of findings highlighting opportunities for future research. The code is available at `https://github.com/timpostuvan/CTDG-link-anomaly-detection`.

## 1 Introduction

Anomaly detection in graphs is the task of identifying abnormal behavior, with applications in spam detection (Ye & Akoglu, 2015), sensor fault detection (Gaddam et al., 2020), financial fraud identification (Dou et al., 2020), and cybersecurity (Zipperle et al., 2022; Reha et al., 2023). Although extensive literature exists on anomaly detection in static graphs (Peng et al., 2018; Ding et al., 2019; Fan et al., 2020; Bandyopadhyay et al., 2019; Yuan et al., 2021; Xu et al., 2022), most real-world networks evolve over time, requiring a focus on dynamic graphs. The most general dynamic graph formulation is the continuous-time dynamic graph (CTDG), where interactions occur at irregular times and carry information as edge attributes.

---

[*]Work done while at Huawei Technologies.
[†]Corresponding author.

Despite the prevalence of dynamic networks, most learning-based graph anomaly detection methods operate on static or discrete-time representations (Goyal et al., 2018; Zheng et al., 2019; Cai et al., 2021), limiting their practical applicability (Rossi et al., 2020; Souza et al., 2022). Other research addresses streaming anomaly detection in dynamic graphs using sketch-based algorithms (Eswaran et al., 2018; Bhatia et al., 2020; 2022), which excel at detecting strong cumulative anomalies (e.g., bursts of edges) rather than individual link-level anomalies, trading granular detection capabilities for efficiency. Reha et al. (2023) study learning-based link anomaly detection in CTDGs but only considers one method, one dataset and *generic* link anomalies.

While there is limited work on anomaly detection in CTDGs, there also exists no categorization of edge-level anomalies in these graphs. Paired with the lack of public CTDG datasets with labeled anomalies, this makes it difficult to analyze learning methods for abilities to detect typed anomalies. On the other hand, the desire of practitioners for detection methods tailored to the specific anomalies of interest in their usecases (Han et al., 2022) highlights the need for ways to characterize and generate anomalies of specific types in CTDGs. For these reasons, a number of works have proposed categorizations of anomalies in various domains (Zhao et al., 2019; Steinbuss & Böhm, 2021; Han et al., 2022), including anomaly classifications for static graph (Liu et al., 2022) and continuous-time discrete graphs (Ma et al., 2021). While these works build their taxonomies for anomalies on structural and contextual graph properties, they lack the explicit temporal perspective necessary for modeling the continuous-time dynamics of CTDGs.

Addressing these limitations, we present the first structured analysis on learning-based graph representation methods in CTDGs applied to *anomalous* edge detection. We introduce a fine-grained taxonomy of edge anomaly types in CTDGs and methods to generate and inject these anomalies into any dataset. In doing so, we generalize and extend the work on static graphs (Liu et al., 2022; Ma et al., 2021) to include time.

To validate our anomaly categorization and anomaly generation procedure, we propose a process for generating synthetic dynamic graphs that are consistent in time, structure, and context. To address the task of *detecting* typed anomalous in CTDG, we build upon the link prediction literature and *adapt* graph-representation learning methods for typed link *anomaly* detection. We show that the direct application of link prediction results in sub-optimal performance in anomaly detection. We identify and mitigate two shortcomings of the general link prediction approach: To make models context-aware, we devise a modification of existing temporal graph methods to predict the existence of a link *conditioned* on context and time. Further, we improve the training regime using typed edge-level perturbations that improve models prediction capabilities w.r.t. the individual dimensions of time, context, and structure.

We benchmark six existing temporal graph methods, spanning from non-parametric memorization, sequential models, and random walk models to graph convolution methods, on the link anomaly detection task. We assess their detection performance on eight datasets for benchmarking, including synthetic and real graphs while using synthetically generated and organic anomalies. We find that the considerations made on synthetic graphs designed with structural, contextual and temporal consistencies generalize to real-world datasets, validating the soundness of our taxonomy. On the other hand, experiments reveal some real graphs do not exhibit strong consistencies along some of these dimensions. We also find that different learning methods are more suited to capturing different aspects of graph normality, hence detecting different types of anomalies, highlighting the benefit of tailoring the choice of model to the dataset. Finally, we present a list of future directions and research opportunities in the field of anomaly detection for CTDG.

In this paper, we make the following contributions:

- We present the first comprehensive analysis of link anomaly detection for continuous-time dynamic graphs.

- We introduce a novel taxonomy for edge-level anomalies in continuous-time dynamic graphs, along with corresponding procedures for generating and injecting typed anomalies (§3.1 and §3.2). We also propose a synthetic graph generation process, where individual anomaly types are distinguishable (§3.3).

- We propose two steps to adapt graph-representation learning methods to the more subtle task of link anomaly detection (§4.2 and §4.3).

- We assess state-of-the-art temporal graph learning methods for detecting typed synthetic and organic anomalies in synthetic and real graphs (§5), highlight future research opportunities (§6).

## 2 Problem Definition

We address the problems of categorizing and detecting anomalous edges in dynamic graphs. A continuous-time dynamic graph (CTDG) extends the concept of a static graph by incorporating timestamps and attributes (also referred to as message) on edges:

**Definition 1** *(**Continuous-time dynamic graph**) A continuous-time dynamic graph is a sequence of non-decreasing chronological interactions $\mathcal{G} = \{(u_1, v_1, t_1, \boldsymbol{m}_1), (u_2, v_2, t_2, \boldsymbol{m}_2), \dots\}$, with $0 \leq t_1 \leq t_2 \leq \dots$, where $u_i, v_i \in \mathcal{V}$ denote the source node and destination node of the $i^{th}$ edge at timestamp $t_i$ and $\mathcal{V}$ is the set of all nodes. Each interaction $(u, v, t, \boldsymbol{m})$ has edge message features $\boldsymbol{m} \in \mathbb{R}^D$, where $D$ is the number of attributes. For non-attributed graphs, edge features are often set to zero vectors, i.e., $\boldsymbol{m} = \boldsymbol{0}$.*

We define the task of link anomaly detection in CTDGs as follows:

**Definition 2** *(**Link anomaly detection in CTDGs**) Given a continuous-time dynamic graph $\mathcal{G}$, the goal of link anomaly detection is to learn a function: $f : \mathcal{V} \times \mathcal{V} \times \mathbb{R} \times \mathbb{R}^D \to \mathbb{R}$ that maps a node pair $(u, v)$, timestamp $t$, and edge message $\boldsymbol{m}$ to a real-valued anomaly score for the edge $e = (u, v, t, \boldsymbol{m})$.*

## 3 Anomaly Taxonomy and Synthesis

**Background**. The categorization and synthesis of realistic anomalies are fundamental for designing, analyzing, and benchmarking anomaly detection methods across domains (Zhao et al., 2019; Steinbuss & Böhm, 2021; Han et al., 2022; Liu et al., 2022; Ma et al., 2021), especially when labeled data is scarce or costly (Han et al., 2022). For static graphs, Liu et al. (2022) introduce a taxonomy and generation method for *structural* and *contextual* anomalies. Ma et al. (2021) briefly discuss another categorization of anomalies in static graphs and re-use it for *discrete-time* dynamic graphs, i.e., graphs represented as timestamped snapshots, while disregarding the effect of time. To date, there exists no taxonomy for anomalies in CTDGs.

**Challenges in CTDG taxonomies**. We identify three challenges when expanding related work from static and discrete-time dynamic graphs to the CTDG domain:

1. With the added temporal dimension, edge and node *histories* co-define "normal" graph behavior, which constantly *evolves* over time (Liu et al., 2020). Such evolving normality is not captured by definitions of anomalies in static graphs (Liu et al., 2022).

2. Recent categorizations such as the following ones from Liu et al. (2022) and Ma et al. (2021) either lack completeness or are too general for our aim:
   - *structural anomalies*, defined as anomalously densely connected nodes in static graphs (Liu et al., 2022), disregard structural anomalies relative to the graphs' history as well as the aspect of communities;
   - *contextual anomalies*, defined for static graphs as nodes with attributes different to those of their neighbors (Liu et al., 2022), are limited to *local* anomalies;
   - The definition of *anomalous edges* for static and discrete-time dynamic graphs (i.e., graph snapshots) (Ma et al., 2021) regards only structural properties explicitly, but not edge-specific histories or attributes. Moreover, this broad definition does not inform about the nature of the edge anomaly.

3. There exist neither real-world nor synthetic CTDGs suitable to validate a fine-grained taxonomy for anomalies in CTDGs.

Here, we propose a fine-grain taxonomy for edge-level anomalies in CTDGs by re-interpreting *context* and *structure*, and incorporating *time* explicitly. We then describe procedures to generate into graphs edges that are anomalous w.r.t. one or multiple of these dimensions. Lastly, we introduce a graph generation process for CTDGs that assures reliable and largely independent consistencies along the structural, temporal, and contextual dimensions. This process is used in §5 to: validate our proposed taxonomy and procedures for synthesizing and injecting edge-level anomalies, and comparatively assess the capabilities of link prediction approaches in detecting the five anomaly types defined by our taxonomy.

Table 1: The proposed taxonomy of synthetic anomaly types with their acronyms. We exemplify each anomaly type using a scenario of a package distribution network, where nodes are either warehouses or customers, and an edge between a warehouse and a customer marks the delivery of a package.

| Anomaly type | Acronym | Example |
|---|---|---|
| Benign | | The ordered package is delivered to the customer on time. |
| Temporal | T | The ordered package is delivered to the customer with delay. |
| Contextual | C | A package with wrong content is delivered to the customer on time. |
| Temporal-contextual | T-C | A package with wrong content is delivered to the customer with delay. |
| Structural-contextual | S-C | The ordered package is delivered to a wrong address on time. |
| Temporal-structural-contextual | T-S-C | A random package is delivered to a random address at a random time. |

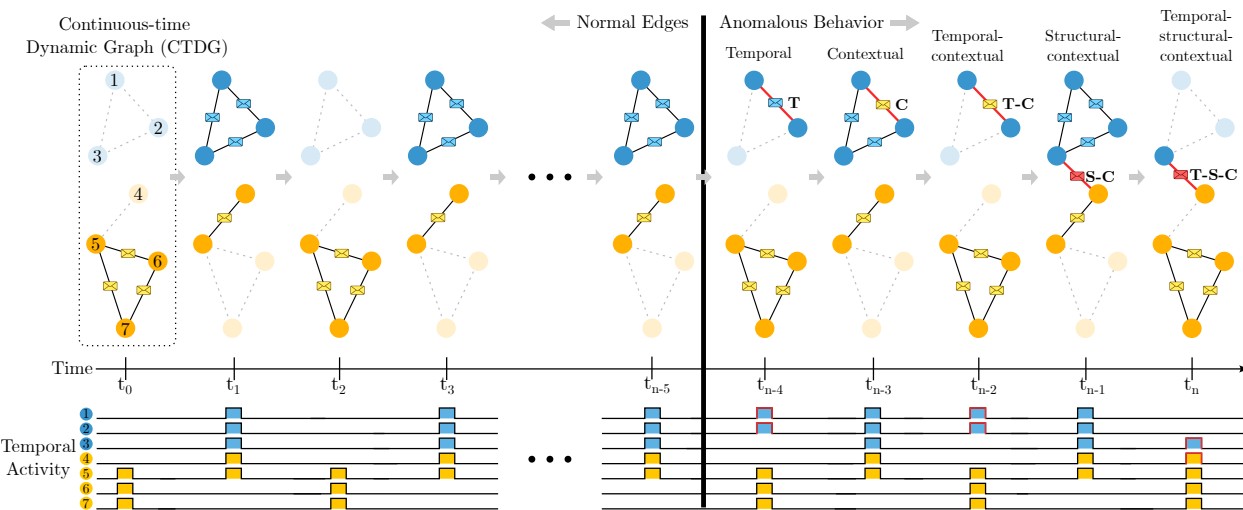

Figure 1: A CTDG with the proposed synthetic anomaly types, visualized in snapshots for clarity. Timestamps $t_0$ to $t_{n-5}$ represent the normal graph behavior, characterized by the properties discussed in §3.1: (1) two node communities define the *structure*, i.e., ●, ●, (2) there are two types of message *contexts* ⊠, ⊠ tied to these communities, (3) the *timing* of edges follows the Temporal Activity at the bottom, i.e., ⬜, ⬜ indicate the node is active at this time. From timestamp $t_{n-4}$ to $t_n$ we highlight and categorize edge anomalous behavior. (T): Edge appears at an unexpected time. (C): Edge message differs from the expected message between the two nodes. (T-C): Edge appears at an unexpected time and its message differs from the expected message between the two nodes. (S-C): Edge connects two nodes that are expected to be active at this time but do not usually connect with each other, its message is out-of-distribution (⊠). (T-S-C): Edge connects two nodes that do not usually connect with each other, the nodes are not expected to be active at this time and its message is out-of-distribution. Icons ⬜, ⬜ mark temporal anomalies.

## 3.1 A Taxonomy for Anomaly Types in CTDGs

Table 1 shows our proposed taxonomy for edge-level anomalies in continuous-time dynamic graphs. The individual anomaly types are described in the following.

**Context**. For CTDGs, we refer to edge attributes as *context*. The predictability of attributes in many dynamic graphs motivates the individual consideration of corresponding anomalies. Contextual normality may be learned from different distributions of edges: the node pair's historical edges, their neighbors, communities, and all available historical edges (Liu et al., 2022).

**Definition 3** (*Contextual Anomaly* (C)) *In a CTDG, a contextual anomaly is an edge whose attributes (message) significantly differ from their expected values.*

**Structure**. Graph *structure*, sometimes referred to as the graph's topology (Jin et al., 2022), describes the interconnectivity of nodes through edges. Common structures include the multipartite structure, where no two endpoints have the same class (color), and the *community structure* where nodes belonging to the same community are more likely to connect. For example, in collaboration or citation networks, nodes (researchers) are more likely to connect (co-author a paper) with other researchers working in the same research area (from the same community) (Ma et al., 2021). Other real-world dynamic graphs such as proximity networks (Poursafaei et al., 2022), political networks (Poursafaei et al., 2022), and interaction networks (Panzarasa et al., 2009; Poursafaei et al., 2022) incorporate similar structural properties.

**Definition 4** *(**Structural Anomaly** (s)) In a CTDG, a structural anomaly is an edge connecting two nodes that are structurally unexpected to connect.*

**Time**. In dynamic graphs, nodes may interconnect following specific *temporal* patterns. These patterns create dynamics where nodes tend to communicate at predictable times, e.g., due to periodicity, other kinds of regularity, or periods of high activity. For example, traffic in industrial areas occurs most likely during rush hours, friends exchange private messages most likely outside of working hours (Panzarasa et al., 2009), and corporate e-mails may be sent most likely between 9-10 am (Shetty & Adibi, 2004). Other networks with temporal consistencies include flights (Huang et al., 2023a), transactions (Huang et al., 2023a), and interactions (Kumar et al., 2019; Nadiri & Takes, 2022).

**Definition 5** *(**Temporal Anomaly** (T)) In a CTDG, a temporal anomaly is an edge that appears at an unexpected time.*

**Combinations of anomaly types**. The three dimensions of CTDGs, i.e., structure, context, and time, are mutually complementary yet not completely orthogonal, meaning not all combinations of anomalies can be considered independently. While an edge can be anomalous w.r.t. time, context, or *both* time *and* context, we argue that in CTDGs, structural anomalies imply contextual anomalies: When an edge connects two nodes that never connected before and are otherwise structurally unlikely to connect, this implies a contextual anomaly, as there exists no meaningful distribution of edges messages to learn contextual normality from. Overall, the following five out of seven possible combinations of anomalies are considered in our taxonomy: T, C, T-C, S-C, T-S-C. Table 1 and Figure 1 exemplifies and illustrates these categorical anomalies, respectively.

### 3.2 Generation of Anomalies

Here, we present efficient strategies to synthesize – for a given CTDG – anomalous edges that fall into the five categories of anomalies defined above. Anomalies of categories T, C, T-C, and S-C are generated by: (1) sampling a benign reference edge, and (2) applying category-specific perturbations to create abnormal properties; Only T-S-C anomalies do not necessitate benign sampling. In compliance with Def. 3-5, we use randomness as a proxy for *abnormality*. For a given observed edge, we assume that: (i) randomizing the destination node creates a structural anomaly, (ii) randomizing the attributes creates a contextual anomaly, and (iii) randomizing the edge's timestamp creates a temporal anomaly. Note that random sampling is a highly efficient and common approach to generating anomalies yet it might produce non-anomalous samples at a negligible error rate. We refer to $e = (u, v, t, \boldsymbol{m})$ as the sampled reference benign edge, with its source $u$, destination $v$, timestamp $t$, and attributes $\boldsymbol{m}$. We inject anomalies as follows:

- (T): create $\hat{e} = (u, v, \hat{t}, \boldsymbol{m})$, where $\hat{t}$ is a randomly selected timestamp within the time span of the data.

- (C): create $\hat{e} = (u, v, t, \hat{\boldsymbol{m}})$ where $\hat{\boldsymbol{m}}$ is the result of $\hat{\boldsymbol{m}} = \mathrm{argmax}_{\boldsymbol{m_j}} \, d(\boldsymbol{m}, \boldsymbol{m_j})$ with $\boldsymbol{m_j} \in \{\boldsymbol{m_0}, ..., \boldsymbol{m_K}\}$, where $d$ is a pair-wise distance metric between multi-dimensional vectors, and $\boldsymbol{m_0}, ..., \boldsymbol{m_K}$ are $K$ randomly sampled edge messages from the entire message set.

- (T-C): create $\hat{e} = (u, v, \hat{t}, \hat{\boldsymbol{m}})$ where $\hat{t}$ and $\hat{\boldsymbol{m}}$ use the above procedures for T and C anomalies.

- (S-C): sample a second benign reference edge $e_2 = (r, q, t', \boldsymbol{m}')$ within a temporal window of size $W$ (i.e., one of $W$ temporally closest edges). Create an edge $\hat{e} = (u, q, t, \hat{\boldsymbol{m}})$ where $q$ is the destination of $e_2$ and $\hat{\boldsymbol{m}}$ is selected with the procedure for injection of contextual anomalies above.

- (T-S-C): create $\hat{e} = (\hat{u}, \hat{v}, \hat{t}, \hat{\boldsymbol{m}})$ where $\hat{u}, \hat{v}$ are randomly sampled nodes, $\hat{t}$ is a randomly sampled timestamp, and $\hat{\boldsymbol{m}}$ is a randomly sampled edge message.

### 3.3 Synthetic Dynamic Graph Generation

We introduce a synthetic CTDG generation process with clear temporal, structural, and contextual consistencies. The generation starts by randomly creating a static graph from a node set $\mathcal{V}$ and a number of temporal edges $E$. Temporal edges are then sampled from existing static edges, ensuring consistency in context, structure, and time. Algorithm 1 outlines the complete process. To enforce consistencies in the generated CTDG, we do the following:

**Structural consistency:** We generate the initial static graph with the Stochastic Block Model (Abbe, 2018), using $M$ predefined communities (Line 1), ensuring structure as randomly sampled nodes likely belong to different communities.

**Contextual consistency:** We uniformly assign nodes to a fixed number of classes $C$, and create expected edge message attributes $\mu_{j,k}^{(m)}$ between class pairs (Lines 2-3). Expected messages are orthogonal for perfect separation, so randomly sampled edge attributes likely differ from these expectations.

**Temporal consistency:** We uniformly sample time windows for each edge's activity (Line 8). For the same edge, we ensure uniform time intervals between its multiple occurrences (Line 10). Randomly sampled timestamps are unlikely to match these predefined windows.

**Generation Process**. The graph is generated iteratively: for each static edge $(u, v)$ in the STATICGRAPH output, a number of temporal edge occurrences $o_e$ is sampled from a Poisson distribution. Then, an active time window of activity $[t_0^e, t_0^e + \Delta t^e]$ is sampled based on a Lognormal distribution and the maximum time-span of the temporal graph $t_{\text{MAX}}$. Then each edge occurrence's timestamp is evenly spaced within the active time window (within some Gaussian noise), and its message is sampled from a Multivariate Normal distribution based on the nodes $u$ and $v$ respective classes. See further details and default parameters in App. A.1. The graph consistencies allow us to use random sampling (along structure, time, or context) to generate anomalies that fall under *exactly one* of the categories in §3.1. We use Alg. 1 to generate a CTDG for experiments in §5.

## 4 Link Anomaly Detection

### 4.1 Link Prediction vs. Link Anomaly Detection

A prominent learning task in CTDGs is link prediction (Nguyen et al., 2018; Kumar et al., 2019; Rossi et al., 2020; Yu et al., 2023; Huang et al., 2023a). While such task resembles that of link anomaly detection in Def. 2, there are significant differences that we outline here.

**Link prediction setting**. When learning parametric methods for link prediction, it is common to assume that the observed edges in $\mathcal{G}$ are legitimate, i.e., they represent normal graph behavior. In this setting, models can be trained with a self-supervised loss function based on maximizing the probability of occurrence of observed edges $e \in \mathcal{G}$ (i.e., positive edges), while minimizing the probability of occurrence of unseen edges $\bar{e} \in \bar{\mathcal{G}}$ (i.e., negative edges). To create the negative set $\bar{\mathcal{G}}$, one uses *random negative sampling*: take all sources of the positive edges in a batch and re-wire them to randomly sampled destinations. Doing so, exactly *one* negative edge $\bar{e}$ is sampled for each positive edge $e = (u, v, t, \boldsymbol{m})$ (Huang et al., 2023a; Poursafaei et al., 2022), where the negative one only differs in the destination:

$$\bar{e} = (u, \bar{z}, t, \boldsymbol{m}) \quad \text{with } \bar{z} \sim \mathcal{U}(\mathcal{V}) \tag{1}$$

The edge pairs $e, \bar{e}$ are used to solve the problem optimizing $f_\theta$ with a binary loss such as cross-entropy:

$$\mathcal{L}(e) + \mathcal{L}(\bar{e}) = \log f_\theta((u, v, t)) + \log(\mathbf{1} - f_\theta((u, \bar{z}, t))).^1 \tag{2}$$

---

[1]The optimization still uses edge attributes $\boldsymbol{m}$ of past neighbors of $u, v$ through Eq. 3.

---

**Algorithm 1:** Synthetic Dynamic Graph Generation

---

**Input** : Nodeset $\mathcal{V}$, n. temporal edges $E$, n. of communities $M$, avg. n. of edge occurrences $\lambda_{\text{occ}}$, graph time span $t_{\text{MAX}}$, mean $\mu_{ts}$ and st.d. $\sigma_{ts}$ for sampling edge time spans, st.d. of timestamps perturbations $\sigma^{(t)}$, n. of node classes $C$, edge message dimension $D$, st.d. for messages perturbations $\sigma^{(m)}$.

**Output:** Synthetic temporal graph $\mathcal{G}$.

---

**1** $\mathcal{E}_s \leftarrow \text{STATICGRAPH}(|\mathcal{V}|, \frac{E}{\lambda_{\text{occ}}}, M)$

**2** $\mathcal{C} \leftarrow \{c_{u_1}, c_{u_2}, \ldots\}$ where $c_{u_i} \sim \mathcal{U}\{1, 2, \ldots, C\}$

**3** $\mu_{j,k}^{(m)} \leftarrow \text{EXPECTEDMESSAGE}(j, k, D), \quad \forall_{j,k} \in \{1, \ldots, C\}^2$

**4** $\mathcal{G} \leftarrow \emptyset$

**5** **for** $e = (u, v) \in \mathcal{E}_s$ **do**

**6** $\quad$ $o_e \sim \text{Pois}(\lambda_{\text{occ}})$

**7** $\quad$ $\Delta t^e \sim \text{Lognormal}(\mu_{ts}, \sigma_{ts})$

**8** $\quad$ $t_0^e \sim \mathcal{U}(0, t_{\text{MAX}} - \Delta t^e)$

**9** $\quad$ **for** $i \leftarrow 1$ **to** $o_e$ **do**

**10** $\quad\quad$ $t_i^e \leftarrow t_0^e + (i-1)\frac{\Delta t^e}{o_e - 1} + \epsilon_i^e$, where $\epsilon_i^e \sim \mathcal{N}(0, \sigma^{(t)})$

**11** $\quad\quad$ $\boldsymbol{m} \sim \mathcal{N}(\mu_{j,k}^{(m)}, \boldsymbol{I}_D \sigma^{(m)})$ where $j = c_u, k = c_v$

**12** $\quad\quad$ $\mathcal{G} \leftarrow \mathcal{G} \cup (u, v, t_i^e, \boldsymbol{m})$

**13** $\quad$ **end**

**14** **end**

**15** **return** $\mathcal{G}$

---

To benefit from the graph structure, the network uses a neighbor sampler which stores node neighbors at each time step. For a node $u$, its past neighbors $v$ are used to aggregate information at time $t$:

$$\mathcal{N}_u^t = \{v | (u, v, t^-, \boldsymbol{m}) \in \mathcal{G} \text{ and } t^- < t\}. \tag{3}$$

After training, the trained model $f_\theta$ is able to provide a likelihood of existence for any arbitrary link.

**Link anomaly detection setting**. Certain aspects differ in the task of link anomaly detection (Def. 2). First, (1) while random negative sampling of Eq. 1 models structural anomalies well (§3.1) by sampling random destinations, it disregards anomalies in the temporal and contextual dimension: $t$ and $\boldsymbol{m}$ are never perturbed. This limits the model's ability to capture non-structural abnormalities. Secondly, (2) the message attributes $\boldsymbol{m}$ are informative, which models should now use in the prediction. Unlike in link prediction, the goal is not to predict the existence of an arbitrary unobserved link, but to estimate the likelihood of occurrence of an observed link. Lastly, (3) the assumption that the observed edges are legitimate (i.e., normal) is invalidated at evaluation time,[2] as a portion of edges in the validation and test set of $\mathcal{G}$ can be anomalous but their normality is unknown. The graph can be seen as the union of normal and anomalous edges $\mathcal{G} = \mathcal{G}^{\text{b}} \cup \mathcal{G}^{\text{a}}$. While the problem can still be formulated as a self-supervised learning problem with negative sampling, this leads to anomalous information entering the neighbor sampler $\mathcal{N}_u^t$ of Eq. 3: $\mathcal{N}_u^t = \{v | (u, v, t^-, \boldsymbol{m}) \in \mathcal{G}^{\text{b}} \cup \mathcal{G}^{\text{a}} \text{ and } t^- < t\}$. While the challenges stemming from (3) can not be addressed directly (see §6), we address problems (1) and (2) in the following.

### 4.2 Conditioning on Context

**Problem**. The optimization problem defined for link prediction (Eq. 2) is sub-optimal in link anomaly detection since the information carried in the edge attributes ($\boldsymbol{m}$) can be used for edge scoring.

**Solution**. We reformulate the optimization problem with the inclusion of the edge attributes as follows:

$$\mathcal{L}(e) + \mathcal{L}(\bar{e}) = \log f_\theta((u, v, t, \boldsymbol{m})) + \log(\boldsymbol{1} - f_\theta((u, \bar{z}, t, \boldsymbol{m}))). \tag{4}$$

---

[2]Note that during training the assumption still holds.

In practice, to incorporate the contextual information of an edge $e = (u, v, t, \boldsymbol{m})$, we project the edge attributes with a linear layer and concatenate them to the node embeddings before feeding the concatenation into the link anomaly detection head. Given the temporal node embeddings at time $t$, $\boldsymbol{h}_u^t, \boldsymbol{h}_v^t$, the prediction head receives $\boldsymbol{h}_u^t || \boldsymbol{h}_v^t || \boldsymbol{W}\boldsymbol{m}$ as the input, $\boldsymbol{W} \in \mathbb{R}^{D \times d_2}, d_2 < D$ is learned with the rest of the model parameters. Note that we propose a straightforward yet versatile and model-agnostic method for incorporating contextual information. Alternatively, contextual attributes could be directly integrated into the message-passing process of each model, enabling the generation of context-aware temporal embeddings.

### 4.3 Improved Training Regime

**Problem**. Because the random negative sampling described in Eq. 1 only modifies edges' destinations, the negative edge set only contains structural anomalies but not temporal or contextual ones.

**Solution**. We propose a novel negative sampler to be used in training. The sampler applies one of the following three perturbations to a positive edge $e = (u, v, t, \boldsymbol{m})$ to create a negative edge $\bar{e}$:

$$
\begin{aligned}
\bar{e} &= (u, \bar{z}, t, \boldsymbol{m}) \quad \text{with } \bar{z} \sim \mathcal{U}(\mathcal{V}) \text{ or} \\
\bar{e} &= (u, v, \bar{t}, \boldsymbol{m}) \quad \text{with } \bar{t} \sim \mathcal{U}[t_0, t_{\text{MAX}}] \text{ or} \\
\bar{e} &= (u, v, t, \bar{\boldsymbol{m}}) \quad \text{with } \bar{\boldsymbol{m}} \sim \mathcal{U}\{\boldsymbol{m}_0, \boldsymbol{m}_1, ..., \}.
\end{aligned}
\tag{5}
$$

These perturbations are universal and do not rely on the proposed injection strategies. In the implementation, the latter sampler samples messages $\boldsymbol{m}$ uniformly among the messages in the batch for efficiency reasons.

## 5 Experiments

We design our experimental setup to answer the following questions: **RQ1** (§5.2) How effective are our methods for adapting link prediction methods to the link anomaly detection task? **RQ2** (§5.3) How valid is our taxonomy and generation process of typed anomalies when applied to synthetic and real-world CTDGs and how discernible are these anomalies? **RQ3** (§5.3, §5.4) How do capabilities of different link prediction methods differ at detecting typed anomalies in synthetic and real datasets? **RQ4** (§5.5) How do methods perform at detecting organic anomalies in real datasets?

### 5.1 Experimental Setup

We provide here information about the experimental setup. Complementary details can be found in the appendix: datasets (App. A.1), model descriptions (App. A.2), hyper-parameter configurations (App. A.3), runtimes (App. B), and extensive prediction results (App. C).

**Datasets**. We run experiments on eight different datasets. The first one is TempSynthGraph – a CTDG we generated using Alg. 1 with 10,000 nodes (see §3.3). We use five benign real-world datasets: Wikipedia, Reddit, MOOC, Enron, and UCI from the work of Poursafaei et al. (2022). Finally, we experiment on two CTDGs from the cybersecurity domain with labeled organic anomalies: LANL (Kent, 2016) and Darpa-Theia (Reha et al., 2023).

**Methods**. We compare seven representative CTDGs learning methods that are based on graph convolutions, memory networks, random walks, and sequential models: TGN (Rossi et al., 2020), CAWN (Wang et al., 2020), TCL (Wang et al., 2021), GraphMixer (Cong et al., 2022), DyGFormer (Yu et al., 2023), and two variants of EdgeBank (Poursafaei et al., 2022), namely EdgeBank$_\infty$ and EdgeBank$_{\text{tw}}$.

**Experimental setting**. We train models for the task of link anomaly detection of Def. 2. Following our considerations from §4 we use binary cross-entropy loss as in Eq. 4 and the negative sampler in Eq. 5. For each method, the same MLP prediction head takes the edge (including time and edge attributes) as input and predicts the probability $p(e)$ of the link's existence. The link anomaly score is then computed as $1-p(e)$. We use the Area Under the Receiver Operating Characteristic curve (AUC) as the main evaluation metric, but we also report Average Precision (AP) and Recall@k metrics in App. C. In datasets with synthetic anomalies,

we chronologically divide each dataset into training (70%), validation (15%), and testing (15%). For the two datasets with organic anomalies LANL and DARPA-THEIA, we use all edges that occurred before the first anomalous edge for training, while the remaining data is split so that validation and testing sets contain an equal number of anomalous edges. For testing, we select models based on their AUC on the validation set.

**Synthetic anomaly injection**. In TEMPSYNTHGRAPH, WIKIPEDIA, REDDIT, UCI, ENRON and MOOC, we inject 5% of synthetic anomalies in the validation and test splits with the procedure described in §3.2. Reference benign edges are sampled within the same data split and with probabilities proportional to $o_e^{-0.5}$, with $o_e$ representing the number of occurrences of edge $e$ in that split. For C, T-C, S-C, and T-S-C anomalies, following Huang et al. (2023b), we adopt cosine similarity as the comparison metric ($d$ in §3.2) with $K=10$ randomly sampled edge messages; For S-C anomalies, $W=20$.

**Implementation details**. We optimize parametric models with Adam (Kingma & Ba, 2015), for 50 epochs, with an early stopping patience of 20. We perform a grid search for learning rate in the range $[10^{-3}, 10^{-6}]$. Batch size is always 200 except in EdgeBank$_\infty$ and EdgeBank$_\text{tw}$, where we set it to 1 to prevent inaccurate look-ups within the same batch. Models have $\sim$100k trainable parameters, with hyper-parameters adapted from Yu et al. (2023). We repeat each experimental run with three weight initializations, with the exception of DARPA-THEIA where we only use one initialization due to compute limitation. When introducing synthetic anomalies, we also average results across three randomly seeded sets of injected anomalies.

## 5.2 Towards Link Anomaly Detection

**Result - Conditioning on context**. To demonstrate the efficacy of the contextual conditioning (see §4.2), we show that no link prediction method trained with Eq. 2 can detect contextual anomalies without contextual conditioning, while the revised optimization of Eq. 4 enables them to do so. We use TEMPSYNTHGRAPH and compute the AUC performance of anomaly detection for contextual anomalies only when training TGN with Eq. 2 vs Eq. 4; we report the result in Tab. 2. The table shows that contextual anomalies go undetected without conditioning on contextual information.

**Result - Improved training regime**. To demonstrate the efficacy of the improved negative sampler (see §4.3), we show that the revised sampler of Eq. 5 encourages the model to learn an enhanced representation of structure, time, and context compared to a random negative sampler (Eq. 1). We use TEMPSYNTHGRAPH and compute the AUC performance of anomaly detection on various anomaly types for models trained with the negative

Table 2: AUC of TGN trained under with and without conditioning on context and improved training regime on TEMPSYNTHGRAPH.

| Anomaly | Conditioning on context | | Improved training | |
| --- | --- | --- | --- | --- |
| | w/o (Eq. 2) | w/ (Eq. 4) | w/o (Eq. 1) | w/ (Eq. 5) |
| T | — | — | $89.29_{\pm 1.92}$ | $90.04_{\pm 1.94}$ |
| C | $52.05_{\pm 0.40}$ | $97.24_{\pm 0.22}$ | $66.49_{\pm 11.14}$ | $97.24_{\pm 0.22}$ |
| T-C | — | — | $89.51_{\pm 1.60}$ | $97.61_{\pm 0.17}$ |
| S-C | — | — | $97.67_{\pm 0.20}$ | $97.83_{\pm 0.60}$ |
| T-S-C | — | — | $97.36_{\pm 0.64}$ | $98.24_{\pm 0.22}$ |

sampler of Eq. 1 vs. Eq. 5; we report the result in Tab. 2. The table shows that the improved negative sampler boosts detection performance on all types of anomalies.

## 5.3 Synthetic Graphs, Synthetic Anomalies

Tab. 3 reports the link anomaly detection AUC on TEMPSYNTHGRAPH with synthetic anomalies. Our key findings are as follows.

**Anomalies on multiple dimensions are easier to detect**. Anomalies that deviate from the norm in multiple respects are more easily identifiable. For example, Tab. 3 shows consistently that T-C anomalies are easier to detect compared to C and T anomalies. Similarly, S-C anomalies are less detectable than T-S-C anomalies. All results in Tab. 3 support this finding within the confidence intervals, corroborating the

Table 3: AUC for all models on TEMPSYNTHGRAPH with synthetic anomalies of the five categories introduced in §3.1. **Bold** and underlined values mark first- and second-best performance across anomaly types.

| Anomaly type | EdgeBank$_\infty$ | EdgeBank$_{tw}$ | TGN | CAWN | TCL | GraphMixer | DyGFormer |
|---|---|---|---|---|---|---|---|
| T | $60.86_{\pm0.18}$ | $60.87_{\pm0.17}$ | $90.04_{\pm1.94}$ | $87.91_{\pm2.06}$ | $\underline{94.32}_{\pm0.90}$ | $\mathbf{96.21}_{\pm0.25}$ | $85.18_{\pm0.71}$ |
| C | $49.52_{\pm0.06}$ | $49.52_{\pm0.06}$ | $97.24_{\pm0.22}$ | $90.92_{\pm8.93}$ | $92.49_{\pm1.91}$ | $\underline{98.32}_{\pm0.11}$ | $\mathbf{98.52}_{\pm0.16}$ |
| T-C | $61.02_{\pm0.12}$ | $61.02_{\pm0.11}$ | $97.61_{\pm0.17}$ | $94.86_{\pm2.34}$ | $96.80_{\pm0.82}$ | $\mathbf{98.89}_{\pm0.05}$ | $\underline{97.83}_{\pm0.23}$ |
| S-C | $98.43_{\pm0.04}$ | $98.43_{\pm0.04}$ | $97.83_{\pm0.60}$ | $\underline{99.03}_{\pm0.03}$ | $97.75_{\pm0.36}$ | $98.51_{\pm0.08}$ | $\mathbf{99.15}_{\pm0.10}$ |
| T-S-C | $\mathbf{99.03}_{\pm0.00}$ | $\mathbf{99.03}_{\pm0.00}$ | $98.24_{\pm0.22}$ | $\mathbf{99.03}_{\pm0.01}$ | $98.38_{\pm0.18}$ | $99.02_{\pm0.01}$ | $\mathbf{99.03}_{\pm0.01}$ |

validity of the graph generation process of §3.3: in TEMPSYNTHGRAPH, edges that are abnormal in multiple dimensions become more anomalous with respect to the normal behavior.

**GraphMixer excels in modeling temporal dynamics**. GraphMixer demonstrates superior performance in detecting T and T-C anomalies on TEMPSYNTHGRAPH. As temporal anomalies differ from their benign counterparts solely through timestamp variations (see §3.2), models require a robust representation of time to better detect them. We note that GraphMixer employs a fixed time-encoding function (Cong et al., 2022), differently than other methods which use learnable ones (Rossi et al., 2020; Wang et al., 2021; 2020). In alignment with findings from Cong et al. (2022), fixed time-encoding functions improve performance by enhancing the stability of training and leading to better modeling of time.

**DyGFormer and CAWN are superior at modeling structure**. DyGFormer and CAWN are well suited for identifying structural anomalies, i.e., S-C and T-S-C, achieving AUCs>99. CAWN is the only method that considers two-hop neighbors to generate node embeddings, suggesting that broader receptive fields increase methods' ability to model structure. On the other hand, DyGFormer considers only one-hop neighbors, but it features a neighbor co-occurrence encoding scheme (Yu et al., 2023) capturing correlations between node pairs and yielding weak structural information about two-hop paths between the nodes of interest. This additional information contributes to DyGFormer's superior performance in detecting structural anomalies.

## 5.4 Real Graphs, Synthetic Anomalies

We report link anomaly detection AUC scores for models on WIKIPEDIA, REDDIT, MOOC, ENRON, and UCI with synthetic anomalies in Tab. 4. Below are the key findings from the table.

**Findings on TempSynthGraph generalize to real graphs with synthetic anomalies**. As in Tab. 3, no single temporal graph method universally outperforms others across all datasets and anomaly types: even on a single anomaly type, no method consistently ranks first in Tab. 4 across all datasets. Similarly, GraphMixer demonstrates superior performance in detecting temporal anomalies also on real graphs, while DyGFormer and CAWN exhibit superior performance in identifying structural anomalies.

**Real graphs may not exhibit strong consistencies**. On TEMPSYNTHGRAPH, best-performing models consistently achieve AUCs>96 (Tab. 3) on each anomaly type, demonstrating the ability of models to accurately identify anomalies in graphs with clear consistencies. Conversely, in real graphs, the best-performing models often exhibit significantly lower detection rates (Tab. 4), indicating a lack of consistencies along the three dimensions. Detection of T anomalies in REDDIT is notably poor, with CAWN achieving the highest result at AUC=81.12. This implies that REDDIT lacks regular temporal patterns, i.e., Reddit users tend to post at irregular times. In WIKIPEDIA, no method detects C anomalies satisfactorily, with the best value being at AUC=84.03, highlighting a lack of consistent contextual patterns. Note that edge attributes in WIKIPEDIA are a low-dimensional representation of the text contained in the page edit, suggesting that these edits carry limited contextual information. In UCI and ENRON, detection of structural anomalies is challenging, with the top two-performing models yielding AUC=86.45 and AUC=88.18, respectively. This suggests, counter-intuitively, that in the communication networks participants may tend to communicate without following strict structural rules. MOOC seems to be the most consistent network across all three aspects, with all best-performing methods scoring with AUCs>96. These findings highlight the irregular

Table 4: AUC for models on Wikipedia, Reddit, MOOC, Enron and UCI with typed synthetic anomalies (§3.2). **Bold** and underlined values mark first- and second-best performing models across rows. We do not report results on Enron and UCI for c and t-c anomalies as these datasets lack edge attributes.

| Anomaly type | Datasets | EdgeBank$_\infty$ | EdgeBank$_{tw}$ | TGN | CAWN | TCL | GraphMixer | DyGFormer |
|---|---|---|---|---|---|---|---|---|
| T | Wikipedia | $58.40_{\pm0.20}$ | $60.29_{\pm0.27}$ | $85.18_{\pm0.47}$ | $\underline{86.31}_{\pm0.36}$ | $84.54_{\pm0.64}$ | $84.55_{\pm0.20}$ | $\mathbf{86.63}_{\pm0.31}$ |
| | Reddit | $55.31_{\pm0.25}$ | $58.96_{\pm0.16}$ | $76.93_{\pm0.38}$ | $\mathbf{81.12}_{\pm0.27}$ | $\underline{77.98}_{\pm0.43}$ | $76.62_{\pm0.39}$ | $76.16_{\pm1.88}$ |
| | MOOC | $50.73_{\pm0.13}$ | $50.53_{\pm0.26}$ | $\mathbf{96.18}_{\pm0.15}$ | $\underline{95.95}_{\pm0.58}$ | $95.71_{\pm0.21}$ | $95.13_{\pm0.40}$ | $94.42_{\pm0.50}$ |
| | Enron | $53.66_{\pm0.11}$ | $54.85_{\pm0.18}$ | $84.19_{\pm2.57}$ | $\underline{89.67}_{\pm1.04}$ | $88.06_{\pm2.15}$ | $\mathbf{91.92}_{\pm0.48}$ | $74.80_{\pm3.40}$ |
| | UCI | $51.97_{\pm1.09}$ | $51.38_{\pm1.17}$ | $\underline{85.72}_{\pm1.06}$ | $83.08_{\pm1.46}$ | $83.08_{\pm1.86}$ | $\mathbf{87.73}_{\pm0.76}$ | $80.95_{\pm1.67}$ |
| | Avg. Rank | 6.60 | 6.40 | $\underline{2.60}$ | $\mathbf{2.20}$ | 3.20 | 2.80 | 4.20 |
| C | Wikipedia | $56.48_{\pm0.06}$ | $58.07_{\pm0.14}$ | $79.67_{\pm0.64}$ | $\underline{84.03}_{\pm0.44}$ | $79.10_{\pm0.60}$ | $79.09_{\pm0.46}$ | $\mathbf{84.22}_{\pm0.46}$ |
| | Reddit | $61.23_{\pm0.42}$ | $71.86_{\pm0.24}$ | $93.81_{\pm0.37}$ | $93.33_{\pm1.94}$ | $\mathbf{95.32}_{\pm0.13}$ | $93.94_{\pm0.27}$ | $\underline{94.81}_{\pm0.45}$ |
| | MOOC | $45.31_{\pm0.51}$ | $44.79_{\pm0.65}$ | $\underline{99.36}_{\pm0.08}$ | $98.40_{\pm0.48}$ | $99.20_{\pm0.23}$ | $98.74_{\pm0.89}$ | $\mathbf{99.57}_{\pm0.08}$ |
| | Avg. Rank | 6.67 | 6.33 | 3.00 | 4.00 | $\underline{2.67}$ | 4.00 | $\mathbf{1.33}$ |
| T-C | Wikipedia | $58.26_{\pm0.06}$ | $60.73_{\pm0.13}$ | $85.02_{\pm0.45}$ | $\mathbf{86.29}_{\pm0.39}$ | $84.42_{\pm0.54}$ | $84.85_{\pm0.52}$ | $\underline{86.24}_{\pm0.47}$ |
| | Reddit | $54.82_{\pm0.31}$ | $58.46_{\pm0.26}$ | $93.39_{\pm0.40}$ | $91.96_{\pm2.61}$ | $\mathbf{94.94}_{\pm0.27}$ | $\underline{94.87}_{\pm0.25}$ | $93.75_{\pm0.96}$ |
| | MOOC | $50.37_{\pm0.39}$ | $50.22_{\pm0.54}$ | $99.58_{\pm0.14}$ | $99.34_{\pm0.13}$ | $99.49_{\pm0.28}$ | $\mathbf{99.72}_{\pm0.16}$ | $\underline{99.64}_{\pm0.19}$ |
| | Avg. Rank | 6.67 | 6.33 | 3.33 | 3.67 | 3.33 | $\mathbf{2.33}$ | $\mathbf{2.33}$ |
| S-C | Wikipedia | $93.29_{\pm0.24}$ | $91.08_{\pm0.23}$ | $86.04_{\pm0.51}$ | $\underline{93.59}_{\pm0.65}$ | $73.45_{\pm10.92}$ | $70.12_{\pm0.56}$ | $\mathbf{94.88}_{\pm0.27}$ |
| | Reddit | $94.78_{\pm0.15}$ | $90.85_{\pm0.12}$ | $90.61_{\pm0.83}$ | $\underline{94.79}_{\pm1.14}$ | $93.50_{\pm0.22}$ | $90.98_{\pm0.46}$ | $\mathbf{96.50}_{\pm0.93}$ |
| | MOOC | $70.03_{\pm0.19}$ | $69.49_{\pm0.11}$ | $\underline{87.41}_{\pm0.89}$ | $\mathbf{96.65}_{\pm0.24}$ | $81.50_{\pm0.65}$ | $79.54_{\pm2.31}$ | $86.52_{\pm0.73}$ |
| | Enron | $85.03_{\pm0.79}$ | $85.15_{\pm0.77}$ | $75.37_{\pm4.05}$ | $\underline{86.96}_{\pm0.73}$ | $63.47_{\pm1.45}$ | $63.77_{\pm1.54}$ | $\mathbf{88.18}_{\pm0.95}$ |
| | UCI | $82.23_{\pm0.13}$ | $81.06_{\pm0.14}$ | $63.83_{\pm2.22}$ | $\mathbf{86.45}_{\pm1.42}$ | $60.16_{\pm2.46}$ | $55.10_{\pm1.84}$ | $\underline{84.66}_{\pm1.50}$ |
| | Avg. Rank | 3.80 | 4.80 | 4.80 | $\mathbf{1.60}$ | 5.40 | 6.00 | $\mathbf{1.60}$ |
| T-S-C | Wikipedia | $94.61_{\pm0.00}$ | $92.33_{\pm0.00}$ | $90.40_{\pm1.07}$ | $\underline{97.37}_{\pm0.23}$ | $92.41_{\pm1.43}$ | $97.04_{\pm0.40}$ | $\mathbf{98.89}_{\pm0.19}$ |
| | Reddit | $96.53_{\pm0.00}$ | $92.44_{\pm0.00}$ | $97.42_{\pm0.78}$ | $\mathbf{98.08}_{\pm0.44}$ | $96.42_{\pm0.28}$ | $\mathbf{98.08}_{\pm0.12}$ | $97.95_{\pm0.81}$ |
| | MOOC | $80.37_{\pm0.01}$ | $76.62_{\pm0.02}$ | $\underline{99.85}_{\pm0.11}$ | $99.73_{\pm0.06}$ | $99.77_{\pm0.05}$ | $\mathbf{99.96}_{\pm0.02}$ | $99.82_{\pm0.04}$ |
| | Enron | $\mathbf{97.41}_{\pm0.07}$ | $\underline{97.39}_{\pm0.09}$ | $74.90_{\pm5.51}$ | $94.21_{\pm0.46}$ | $93.06_{\pm0.69}$ | $95.90_{\pm0.42}$ | $91.89_{\pm0.99}$ |
| | UCI | $85.65_{\pm0.05}$ | $84.07_{\pm0.00}$ | $89.62_{\pm2.53}$ | $\underline{94.51}_{\pm1.42}$ | $92.21_{\pm1.26}$ | $93.73_{\pm0.49}$ | $\mathbf{96.11}_{\pm0.63}$ |
| | Avg. Rank | 4.40 | 5.80 | 5.00 | 3.00 | 4.80 | $\mathbf{2.20}$ | $\underline{2.80}$ |

nature of real graphs, supporting the use of synthetic graph generation methods, such as Alg. 1, to assess and compare the capabilities of models in controlled scenarios. We present a comparison of the statistical properties of synthetic and real-world graphs in Appendix D.

## 5.5 Real Graphs, Organic Anomalies

Tab. 5 reports the link anomaly detection AUC on LANL and Darpa-Theia with organic anomalies. Here we describe the main findings.

**Organic anomalies in LANL and Darpa-Theia resemble synthetic structural anomalies**. DyG-Former and CAWN exhibit the best performance on real datasets with organic anomalies, matching their good performance in detecting s-c and t-s-c synthetic anomalies in §5.4. In contrast, EdgeBank$_{tw}$ weakly outperforms a random baseline: it achieves AUC<67 in LANL and AUC<53 in Darpa-Theia. This difference implies that while methods apt at detecting structural anomalies work well, memorization-based approaches such as EdgeBank are not well-suited to detect them. In particular, the extremely high proportion of unseen test edges in Darpa-Theia and LANL requires learning from non-structural aspects of the CTDG (Reha et al., 2023; Poursafaei et al., 2022).

**Significant disparities in model performances**. There are considerable gaps between the top-performing and lowest-performing learning-based models in LANL and Darpa-Theia. This indicates that careful selection of an appropriate method is crucial when dealing with real-world graphs containing organic anomalies: the best learning-based method outperforms the worst one by 6.8 and 21.39 AUC points, respectively.

Table 5: AUC for models on LANL and Darpa-Theia with organic anomalies. **Bold** and underlined values mark first- and second-best performing models across rows.

| Datasets | EdgeBank$_{tw}$ | TGN | CAWN | TCL | GraphMixer | DyGFormer |
|---|---|---|---|---|---|---|
| LANL | 66.84 | $87.54_{\pm 3.16}$ | $\mathbf{94.34}_{\pm 0.06}$ | $89.31_{\pm 1.32}$ | $88.74_{\pm 0.79}$ | $\underline{94.07}_{\pm 3.75}$ |
| Darpa-Theia | 52.72 | 77.18 | 84.18 | 90.30 | 69.03 | **90.42** |
| Avg. Rank | 6.50 | 4.50 | 2.00 | 2.50 | 4.50 | **1.50** |

# 6 Future Directions

We discuss future directions for CTDG link anomaly detection in this section.

**Opportunity 1: New CTDG datasets with organic anomalies**. Previous research highlights the importance of datasets with labeled organic anomalies for method comparison (Liu et al., 2022; Han et al., 2022). While synthetic anomalies are useful, they cannot fully substitute organic ones. In CTDG, only LANL (Kent, 2016) and DARPA (Reha et al., 2023) with organic anomalies exist, both from the cybersecurity domain. The limited and narrow testbed hinders comprehensive method comparison. Thus, developing datasets across diverse domains is essential for robust method evaluation and exploring innovative approaches.

**Opportunity 2: Explainability of methods**.

The ability to explain anomalies is crucial for practitioners (Panjei et al., 2022), especially in safety-critical sectors like healthcare (Ukil et al., 2016), financial services (Ahmed et al., 2016), and self-driving car manufacturing (Bogdoll et al., 2022). While efforts to elucidate temporal graph methods exist (He et al., 2022; Vu & Thai, 2022; Longa et al., 2023), only recently have explainable methods for CTDGs been introduced (Xia et al., 2022). Explainability enhances our understanding of methods and their predictions, providing deeper insights into organic anomalies. Comparing patterns between synthetic and organic anomalies allows drawing parallels, which improves method understanding and highlights distinctive features of organic anomalies.

**Opportunity 3: Reducing the gap between organic and synthetic anomalies**. In §5.5, we observe that certain synthetic anomalies resemble organic ones, such as structural anomalies in LANL and Darpa-Theia. Specific anomaly detection methods excel at detecting particular types (§5.3). These findings support using synthetic anomalies as proxies for model selection, reducing reliance on organic anomalies. This can be achieved by using domain knowledge about prevalent anomaly types or by automating pipelines to map organic anomalies to matching synthetic types. This approach allows selecting detection methods based on synthetic anomaly validation sets, streamlining experimentation.

**Opportunity 4: Expansion of taxonomy in CTDG beyond the edge-level anomalies**.

Anomalies in static graphs are traditionally studied at various levels: edges, nodes, sub-graphs, and entire graphs (Ma et al., 2021). A promising future research direction involves extending our analysis and taxonomy of anomalies in CTDG beyond the edge level.

# 7 Conclusions

We presented a pioneering analysis of link anomaly detection in continuous-time dynamic graphs, addressing a gap in the current literature. Our comprehensive taxonomy of edge anomaly types and corresponding generation strategies, coupled with a synthetic graph generation process, enables better evaluation and understanding of anomaly detection in CTDG. Our experimental findings, together with a discussion on future research opportunities, enable further research in learning anomaly detection methods for CTDGs.

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

# A  Detailed Experimental Settings

We report here details of the datasets, methods and model configurations used in §5.

## A.1  Datasets

In our experiments, we use the proposed synthetic graph (§3.3), five datasets collected by Poursafaei et al. (2022), the Los Alamos National Lab (LANL) dataset (Kent, 2016) and the DARPA-THEIA dataset (Reha et al., 2023). We report an overview of each dataset's statistics in Table 6, and describe them here:

- TEMPSYNTHGRAPH comprises $N = 10,000$ nodes divided into $M = 10$ communities of equal size. The underlying static graph is generated using a stochastic block model (Abbe, 2018), featuring six times more static edges within communities than between them. The temporal graph consists of roughly $E = 1,000,000$ temporal edges, where each unique edge (i.e., an edge between a given pair of nodes) occurs $\lambda = 50$ times on average. The graph spans $t_{MAX} = 10^8$ steps. The parameters for sampling edge time spans, $\mu_{ts}$ and $\sigma_{ts}$, are configured to make edge spans, on average, 1% of the entire graph time span, with a standard deviation of 0.5%. The timestamps of temporal edges are perturbed with $\sigma^{(t)}$ equaling 5% of the average inter-event time of the edge: $\frac{\Delta t^e}{o_e}$. Nodes are assigned to one of $C = 5$ types. The edge message dimension is $D = 100$. The function EXPECTEDMESSAGE$(j, k, D)$ of Alg. 1 returns the mean edge messages $\mu_{j,k}^{(m)}$ where the $l^{\text{th}}$ element is computed as follows:

$$\mu_{j,k}^{(m)}(l) = \begin{cases} 1 & \text{if } \frac{D}{C^2}(C(j-1)+k-1) < l \leq \frac{D}{C^2}(C(j-1)+k) \\ 0 & \text{otherwise,} \end{cases}$$

creating orthogonal mean edge messages, as these are constructed by concatenating blocks of zeros to a single block of ones, so that each class pair has ones in a different location of the vector. In practice,

classes pairs' edge message would look as follows:

$$\mu_{1,1}^{(m)} = [1, 1, 1, 1, 0, 0, 0, 0, ..., 0, 0, 0, 0]$$
$$\mu_{1,2}^{(m)} = [0, 0, 0, 0, 1, 1, 1, 1, ..., 0, 0, 0, 0]$$
$$...$$
$$\mu_{c,c}^{(m)} = [0, 0, 0, 0, 0, 0, 0, 0, ..., 1, 1, 1, 1],$$

the length of a consecutive block of 1s (or 0s) is $\frac{D}{C^2}$. In sampling edge messages, the mean edge messages are perturbed with Gaussian noise with standard deviation $\sigma^{(m)} = 0.05$. Note that orthogonality between mean edge messages is not a general requirement but we use it here to guarantee small overlaps between randomly perturbed edge messages belonging to classes, with the goal of isolating contextual anomalies. Less strict implementations of EXPECTEDMESSAGE can be used in principle.

- WIKIPEDIA (Kumar et al., 2019) is a bipartite interaction graph encompassing edits made to Wikipedia pages within a one-month timeframe. Users and pages are represented as nodes, while links signify editing interactions along with corresponding timestamps. Each link is accompanied by a 172-dimensional Linguistic Inquiry and Word Count feature (Pennebaker et al., 2001) that is derived from the content of the edit.

- REDDIT (Kumar et al., 2019) is a bipartite network that documents user posts under various subreddits over the course of one month. Users and subreddits serve as nodes, and the connections between them represent timestamped posting requests. Each link is accompanied by a 172-dimensional Linguistic Inquiry and Word Count feature (Pennebaker et al., 2001), derived from the content of the respective post.

- MOOC (Kumar et al., 2019) is a bipartite interaction network, consisting of actions done by students in a MOOC online course. Students and course units (e.g., videos and problem sets) are represented as nodes, while a link within the network signifies a student's interaction with a particular content unit and is associated with a 4-dimensional feature.

- ENRON (Shetty & Adibi, 2004) contains email correspondences exchanged among ENRON energy company employees over a three-year period. The dataset has no edge features.

- UCI (Panzarasa et al., 2009) is an online communication network among students of the University of California at Irvine. Nodes represent university students, and links are messages posted by students. The links have no edge features.

- Los Alamos National Lab (LANL) dataset (Kent, 2016) comprises authentication events occurring over 58 consecutive days within the corporate internal computer network of the Los Alamos National Laboratory. Computers are denoted as nodes, and the links between them represent authentication events. Events associated with the red team, which exhibit bad behavior, are labeled as malicious links - organic anomalies in §5. The dataset lacks edge features. We pre-process the LANL dataset according to the methodology outlined in King & Huang (2023), but without combining edges into discrete graph snapshots. Due to the substantial size of the dataset (1.6 billion edges), we evenly subsample the benign edges to obtain approximately 1,000,000 total edges.

- DARPA-THEIA (Reha et al., 2023) is system-level data provenance graphs, constructed from the audit logs collection made by DARPA Engagement 3 (DARPA, 2020). The dataset contains both benign and malicious activities, where the latter are carried out by a red team and represent organic anomalies. Nodes represent files, processes, and sockets, while links correspond to Linux system calls with timestamps. Each link is associated with a 47-dimensional edge feature, describing the system call and the involved nodes.

## A.2 Temporal Graph Learning Models

We select the following representative temporal graph learning models to evaluate them on link anomaly detection task, we also refer the reader to Huang et al. (2023a) for extended descriptions of these methods, which this section borrows from:

Table 6: Statistics of the datasets. All datasets have a time granularity of Unix timestamps.

| Datasets | Domain | #Nodes | #Edges | #Edge features | Bipartite | Duration | #Unique Steps |
|---|---|---|---|---|---|---|---|
| TEMPSYNTHGRAPH | Synthetic | 10,000 | 1,002,325 | 100 | False | $10^8$ steps | 997,256 |
| WIKIPEDIA | Social | 9,227 | 157,474 | 172 | True | 1 month | 152,757 |
| REDDIT | Social | 10,984 | 672,447 | 172 | True | 1 month | 588,915 |
| MOOC | Interaction | 7,144 | 411,749 | 4 | True | 17 months | 345,600 |
| ENRON | Social | 184 | 125,235 | – | False | 3 years | 22,632 |
| UCI | Social | 1,899 | 59,835 | – | False | 196 days | 58,911 |
| LANL (ours) | Cybersecurity | 13,081 | 1,030,147 | – | False | 58 days | 1,027,178 |
| DARPA-THEIA | Cybersecurity | 1,043,224 | 8,416,187 | 47 | False | 247 hours | 1,766,816 |

- *TGN* (Rossi et al., 2020) is a general framework for learning on continuous-time temporal graphs. It keeps a dynamic memory for each node and updates this memory when the node is observed in an event. This is realized by a memory module, a message function, a message aggregator, and a memory updater. To obtain the temporal representation of a node, embedding module aggregates the node's memory and memories of its neighbors.

- *CAWN* (Wang et al., 2020) initially extracts multiple causal anonymous walks for a node of interest. These walks serve to explore the causality within network dynamics and operate on relative node identities, making the model inductive. Subsequently, it employs recurrent neural networks to encode each walk and aggregates the resulting representations to derive the final node representation.

- *TCL* (Wang et al., 2021) utilizes a transformer module to create temporal neighborhood representations for nodes engaged in an interaction. Then, it models the inter-dependencies by employing a co-attentional transformer at a semantic level. To achieve this, TCL incorporates two distinct encoders to derive representations from the temporal neighborhoods surrounding the two nodes forming an edge. Additionally, during the training process, TCL employs a temporal contrastive objective function to preserve the high-level semantics of interactions in latent representations of the nodes.

- *GraphMixer* (Cong et al., 2022) is a simple model architecture that comprises three components: a link-encoder that summarizes information from temporal links with multi-layer perceptions (MLP), a node-encoder that summarizes node information with neighbor mean-pooling, and a link classifier that performs link prediction based on the outputs of the encoders. It is noteworthy that none of these components incorporate graph neural network modules. Additionally, GraphMixer introduced a fixed time-encoding function that encodes any timestamp as an easily distinguishable input vector.

- *DyGFormer* (Yu et al., 2023) is a transformer-based architecture for dynamic graph learning. It first extracts historical first-hop interactions of the nodes participating in the interaction, yielding two interaction sequences. Then, it utilizes a neighbor co-occurrence encoding scheme to model correlations between the two nodes, based on the two interaction sequences. DyGFormer splits the interaction sequences enriched with the neighbor co-occurrence encoding into multiple patches and feeds them into a transformer for capturing long-term temporal dependencies. Finally, the node representations are derived by averaging the corresponding outputs of the transformer.

- *EdgeBank$_\infty$* (Poursafaei et al., 2022) is a simple heuristic that stores all observed edges in a memory. Given an edge, EdgeBank$_\infty$ predicts it as positive if the corresponding node pair is found in the memory, otherwise, EdgeBank$_\infty$ predicts it as negative.

- *EdgeBank$_{\mathrm{tw}}$* (Poursafaei et al., 2022) is a version of EdgeBank that only memorizes edges from a limited time window in the recent past. Therefore, it has a strong recency bias and forgets edges that occurred long ago.

### A.3   Model Configurations

We present the hyper-parameter settings of temporal graph learning models that are fixed across all the datasets. For each method, we always search through the following learning rates: 1e-3, 3e-4, 1e-4, 3e-5,

1e-5, 3e-6, 1e-6. We execute experiments on a cluster with 1000 Intel(R) Xeon(R) CPUs @ ∼2.60GHz. Note that, since experiments for Darpa-Theia would take months under our model configuration and hardware setup, we run EdgeBank$_\infty$ and EdgeBank$_{tw}$ with a batch size of 100 (rather than 1, see §5.1), and models on this dataset are stopped after 25 days of execution time. In practice only TGN does not converge in this timespan. The results for the best checkpoint by the 25$^{th}$ day are reported. The rest of the model configurations are as follows:

- **TGN**:
    - Dimension of time encoding: 50
    - Dimension of node memory: 50
    - Dimension of output representation: 50
    - Number of sampled neighbors: 10
    - Neighbor sampling strategy: recent
    - Number of graph attention heads: 2
    - Number of graph convolution layers: 1
    - Memory updater: gated recurrent unit (Cho et al., 2014)
    - Dropout rate: 0.1

- **CAWN**:
    - Dimension of time encoding: 25
    - Dimension of position encoding: 25
    - Dimension of output representation: 25
    - Number of attention heads for encoding walks: 8
    - Number of random walks: 36
    - Length of each walk (including the target node): 2
    - Time scaling factor $\alpha$: 1e-6
    - Dropout rate: 0.1

- **TCL**:
    - Dimension of time encoding: 50
    - Dimension of depth encoding: 50
    - Dimension of output representation: 50
    - Number of sampled neighbors: 20
    - Neighbor sampling strategy: recent
    - Number of attention heads: 2
    - Number of transformer layers: 2
    - Dropout rate: 0.1

- **GraphMixer**:
    - Dimension of time encoding: 60
    - Dimension of output representation: 60
    - Number of sampled neighbors: 20
    - Neighbor sampling strategy: recent
    - Number of MLP-Mixer layers: 2
    - Time gap: 2000
    - Dropout rate: 0.4

- **DyGFormer**:

- – Dimension of time encoding: 50

- – Dimension of neighbor co-occurrence encoding: 20

- – Dimension of aligned encoding: 20

- – Dimension of output representation: 50

- – Maximal length of input sequences: 32

- – Patch size: 1

- – Neighbor sampling strategy: recent

- – Number of attention heads: 2

- – Number of transformer layers: 1

- – Dropout rate: 0.1

# B Computational Efficiency

Table 7 contains training plus validation time of the temporal graph methods on TEMPSYNTHGRAPH. The most computationally efficient methods in our study are TCL and DyGFormer, both based on the transformers architecture (Vaswani et al., 2017). GraphMixer and TGN exhibit a significantly longer training and validation times, approximately three times slower than the aforementioned methods. CAWN emerges as the slowest among the considered techniques, primarily due to the sampling of random walks at two-hops (compared to the remaining methods being 1-hop).

Table 7: Training and validation times of temporal graph methods per epoch on the synthetic graph. Results are reported in minutes and averaged over 20 runs. Measurements are performed on an Ubuntu machine equipped with one Intel(R) Xeon(R) CPU E5-2658A v3 @ 2.20GHz with 24 physical cores.

|  | TGN | CAWN | TCL | GraphMixer | DyGFormer |
|---|---|---|---|---|---|
| **Time** [min] | $149.29_{\pm 4.21}$ | $173.37_{\pm 15.01}$ | $\mathbf{41.25}_{\pm 3.33}$ | $145.82_{\pm 5.82}$ | $\underline{52.31}_{\pm 1.86}$ |

# C Detailed Experimental Results

Table 8: AP for all models on TEMPSYNTHGRAPH with synthetic anomalies of the five categories introduced in §3.1. **Bold** and underlined values mark first- and second-best performance across anomaly types.

| Anomaly type | EdgeBank$_\infty$ | EdgeBank$_\text{tw}$ | TGN | CAWN | TCL | GraphMixer | DyGFormer |
|---|---|---|---|---|---|---|---|
| T | $16.77_{\pm 0.38}$ | $16.77_{\pm 0.37}$ | $35.64_{\pm 12.78}$ | $37.90_{\pm 4.59}$ | $\underline{53.31}_{\pm 6.95}$ | $\mathbf{64.76}_{\pm 0.91}$ | $37.70_{\pm 1.00}$ |
| C | $4.74_{\pm 0.00}$ | $4.74_{\pm 0.00}$ | $49.81_{\pm 1.86}$ | $44.74_{\pm 10.84}$ | $31.57_{\pm 3.50}$ | $\underline{58.97}_{\pm 4.09}$ | $\mathbf{61.11}_{\pm 3.30}$ |
| T-C | $17.10_{\pm 0.25}$ | $17.11_{\pm 0.24}$ | $\underline{59.08}_{\pm 2.20}$ | $53.68_{\pm 1.80}$ | $54.72_{\pm 4.05}$ | $\mathbf{68.51}_{\pm 1.60}$ | $48.94_{\pm 4.40}$ |
| S-C | $70.98_{\pm 0.07}$ | $70.98_{\pm 0.07}$ | $48.69_{\pm 3.54}$ | $\underline{71.61}_{\pm 0.07}$ | $56.27_{\pm 4.52}$ | $64.72_{\pm 2.09}$ | $\mathbf{77.66}_{\pm 2.09}$ |
| T-S-C | $72.01_{\pm 0.01}$ | $72.02_{\pm 0.00}$ | $60.04_{\pm 3.40}$ | $\underline{72.40}_{\pm 0.28}$ | $64.42_{\pm 2.68}$ | $70.72_{\pm 0.68}$ | $\mathbf{72.48}_{\pm 0.27}$ |

We report the counterparts of the experiments reported in Table 3, 4, 5 results with Average Precision and Recall@k metrics in Table 8, 9, 10, 11 and 12, 13. For Recall@k, $k$ is set to the number of known anomalies for the considered data split.

Table 9: Recall@k for all models on TEMPSYNTHGRAPH with synthetic anomalies of the five categories introduced in §3.1. **Bold** and underlined values mark first- and second-best performance across anomaly types.

| **Anomaly type** | EdgeBank$_\infty$ | EdgeBank$_{tw}$ | TGN | CAWN | TCL | GraphMixer | DyGFormer |
|---|---|---|---|---|---|---|---|
| T | $24.86_{\pm0.38}$ | $24.87_{\pm0.37}$ | $38.83_{\pm13.86}$ | $45.25_{\pm5.66}$ | $\underline{59.98}_{\pm6.27}$ | $\mathbf{69.33}_{\pm0.49}$ | $48.81_{\pm0.57}$ |
| C | $3.61_{\pm0.20}$ | $3.61_{\pm0.20}$ | $48.34_{\pm1.61}$ | $49.15_{\pm8.46}$ | $31.87_{\pm3.75}$ | $\underline{57.64}_{\pm4.08}$ | $\mathbf{60.18}_{\pm3.45}$ |
| T-C | $24.99_{\pm0.36}$ | $24.96_{\pm0.39}$ | $\underline{59.28}_{\pm2.44}$ | $57.18_{\pm3.98}$ | $54.81_{\pm3.69}$ | $\mathbf{67.78}_{\pm1.74}$ | $48.76_{\pm4.35}$ |
| S-C | $70.25_{\pm0.07}$ | $70.25_{\pm0.07}$ | $47.01_{\pm3.23}$ | $\underline{70.53}_{\pm0.54}$ | $55.94_{\pm4.84}$ | $63.53_{\pm2.21}$ | $\mathbf{73.21}_{\pm1.63}$ |
| T-S-C | $70.93_{\pm0.40}$ | $70.83_{\pm0.28}$ | $59.21_{\pm3.45}$ | $\mathbf{72.26}_{\pm0.22}$ | $63.73_{\pm2.92}$ | $70.17_{\pm1.15}$ | $\underline{72.02}_{\pm0.25}$ |

Table 10: AP for models on WIKIPEDIA, REDDIT, MOOC, ENRON and UCI with typed synthetic anomalies (§3.2). **Bold** and underlined values mark first- and second-best performing models across rows. We do not report results on ENRON and UCI for C and T-C anomalies as these datasets lack edge attributes.

| **Anomaly type** | **Datasets** | EdgeBank$_\infty$ | EdgeBank$_{tw}$ | TGN | CAWN | TCL | GraphMixer | DyGFormer |
|---|---|---|---|---|---|---|---|---|
| T | WIKIPEDIA | $6.71_{\pm0.08}$ | $6.95_{\pm0.09}$ | $16.26_{\pm0.57}$ | $\underline{16.97}_{\pm0.38}$ | $14.73_{\pm0.90}$ | $14.27_{\pm0.33}$ | $\mathbf{17.29}_{\pm0.24}$ |
| | REDDIT | $5.98_{\pm0.09}$ | $6.55_{\pm0.05}$ | $11.84_{\pm0.35}$ | $\mathbf{16.06}_{\pm0.38}$ | $\underline{12.59}_{\pm0.61}$ | $12.05_{\pm0.50}$ | $12.47_{\pm1.22}$ |
| | MOOC | $4.83_{\pm0.01}$ | $4.81_{\pm0.02}$ | $\mathbf{64.07}_{\pm1.71}$ | $61.44_{\pm3.59}$ | $57.56_{\pm2.93}$ | $\underline{62.68}_{\pm3.36}$ | $53.81_{\pm4.38}$ |
| | ENRON | $6.03_{\pm0.06}$ | $6.36_{\pm0.10}$ | $15.84_{\pm3.77}$ | $\underline{35.84}_{\pm8.18}$ | $25.05_{\pm6.72}$ | $\mathbf{42.84}_{\pm4.45}$ | $12.09_{\pm3.37}$ |
| | UCI | $4.96_{\pm0.13}$ | $4.90_{\pm0.13}$ | $17.55_{\pm1.63}$ | $\underline{17.88}_{\pm1.71}$ | $15.11_{\pm1.79}$ | $\mathbf{21.91}_{\pm0.93}$ | $15.74_{\pm1.36}$ |
| | Avg. Rank | 6.60 | 6.40 | 3.20 | **2.00** | 3.60 | 2.60 | 3.60 |
| C | WIKIPEDIA | $6.06_{\pm0.02}$ | $6.26_{\pm0.04}$ | $13.91_{\pm0.59}$ | $\mathbf{17.23}_{\pm0.77}$ | $11.99_{\pm0.44}$ | $12.75_{\pm0.44}$ | $\underline{16.57}_{\pm0.41}$ |
| | REDDIT | $9.08_{\pm0.29}$ | $12.48_{\pm0.15}$ | $43.09_{\pm3.19}$ | $38.03_{\pm8.13}$ | $\mathbf{57.25}_{\pm1.51}$ | $46.25_{\pm2.60}$ | $\underline{49.94}_{\pm4.16}$ |
| | MOOC | $4.44_{\pm0.02}$ | $4.40_{\pm0.03}$ | $\underline{92.75}_{\pm1.82}$ | $70.01_{\pm10.36}$ | $85.75_{\pm6.13}$ | $79.30_{\pm6.10}$ | $\mathbf{94.18}_{\pm2.55}$ |
| | Avg. Rank | 6.67 | 6.33 | 3.00 | 3.67 | 3.00 | 3.67 | **1.67** |
| T-C | WIKIPEDIA | $6.66_{\pm0.02}$ | $7.11_{\pm0.05}$ | $15.76_{\pm0.67}$ | $\underline{17.06}_{\pm0.45}$ | $14.43_{\pm0.70}$ | $15.17_{\pm0.49}$ | $\mathbf{17.31}_{\pm0.80}$ |
| | REDDIT | $5.81_{\pm0.11}$ | $6.40_{\pm0.08}$ | $49.94_{\pm3.15}$ | $40.70_{\pm5.46}$ | $\mathbf{55.29}_{\pm4.82}$ | $\underline{50.24}_{\pm2.58}$ | $44.44_{\pm9.09}$ |
| | MOOC | $4.80_{\pm0.04}$ | $4.78_{\pm0.05}$ | $90.54_{\pm3.17}$ | $82.56_{\pm3.31}$ | $85.85_{\pm8.16}$ | $88.85_{\pm2.36}$ | $\mathbf{93.96}_{\pm1.90}$ |
| | Avg. Rank | 6.67 | 6.33 | 2.67 | 4.00 | 3.33 | 3.00 | **2.00** |
| S-C | WIKIPEDIA | $30.43_{\pm0.22}$ | $23.62_{\pm0.17}$ | $21.29_{\pm2.10}$ | $\underline{30.45}_{\pm2.54}$ | $10.33_{\pm2.80}$ | $7.45_{\pm0.20}$ | $\mathbf{30.85}_{\pm1.86}$ |
| | REDDIT | $\underline{39.76}_{\pm0.18}$ | $23.64_{\pm0.09}$ | $31.54_{\pm2.71}$ | $34.49_{\pm6.23}$ | $\mathbf{48.96}_{\pm2.13}$ | $34.00_{\pm2.72}$ | $35.93_{\pm6.19}$ |
| | MOOC | $8.27_{\pm0.05}$ | $7.88_{\pm0.02}$ | $56.60_{\pm7.47}$ | $\mathbf{70.56}_{\pm6.97}$ | $47.97_{\pm8.60}$ | $45.04_{\pm6.43}$ | $\underline{56.78}_{\pm6.63}$ |
| | ENRON | $\mathbf{44.03}_{\pm1.31}$ | $\underline{38.44}_{\pm1.16}$ | $19.09_{\pm6.56}$ | $17.62_{\pm1.30}$ | $6.66_{\pm0.49}$ | $6.99_{\pm0.31}$ | $20.20_{\pm4.69}$ |
| | UCI | $13.34_{\pm0.06}$ | $12.36_{\pm0.05}$ | $6.99_{\pm0.41}$ | $\underline{16.17}_{\pm1.86}$ | $6.23_{\pm0.39}$ | $5.33_{\pm0.35}$ | $\mathbf{16.93}_{\pm1.37}$ |
| | Avg. Rank | 3.00 | 4.80 | 4.60 | 2.80 | 4.80 | 6.00 | **2.00** |
| T-S-C | WIKIPEDIA | $31.70_{\pm0.00}$ | $24.57_{\pm0.00}$ | $31.03_{\pm2.89}$ | $64.79_{\pm5.28}$ | $35.29_{\pm7.09}$ | $\underline{69.28}_{\pm5.87}$ | $\mathbf{84.06}_{\pm2.83}$ |
| | REDDIT | $41.90_{\pm0.00}$ | $24.86_{\pm0.00}$ | $64.67_{\pm5.00}$ | $\mathbf{67.52}_{\pm5.64}$ | $54.74_{\pm3.51}$ | $59.84_{\pm3.41}$ | $\underline{67.22}_{\pm6.25}$ |
| | MOOC | $11.30_{\pm0.00}$ | $9.66_{\pm0.01}$ | $\mathbf{96.36}_{\pm2.23}$ | $95.90_{\pm0.79}$ | $95.77_{\pm1.79}$ | $93.51_{\pm1.94}$ | $94.66_{\pm2.43}$ |
| | ENRON | $\mathbf{63.69}_{\pm0.11}$ | $\underline{56.27}_{\pm0.13}$ | $21.75_{\pm8.29}$ | $49.32_{\pm7.40}$ | $39.35_{\pm4.35}$ | $52.55_{\pm6.95}$ | $27.14_{\pm4.47}$ |
| | UCI | $14.83_{\pm0.02}$ | $13.55_{\pm0.00}$ | $26.03_{\pm9.34}$ | $39.94_{\pm8.54}$ | $\underline{35.11}_{\pm3.26}$ | $33.70_{\pm5.86}$ | $\mathbf{52.72}_{\pm8.21}$ |
| | Avg. Rank | 4.80 | 6.00 | 4.40 | **2.40** | 4.00 | 3.60 | 2.80 |

Table 11: Recall@k for models on WIKIPEDIA, REDDIT, MOOC, ENRON and UCI with typed synthetic anomalies (§3.2). **Bold** and underlined values mark first- and second-best performing models across rows. We do not report results on ENRON and UCI for C and T-C anomalies as these datasets lack edge attributes.

| Anomaly type | Datasets | EdgeBank$_\infty$ | EdgeBank$_{tw}$ | TGN | CAWN | TCL | GraphMixer | DyGFormer |
|---|---|---|---|---|---|---|---|---|
| T | WIKIPEDIA | $13.75_{\pm0.49}$ | $12.39_{\pm0.94}$ | $\mathbf{18.64}_{\pm1.01}$ | $\underline{18.53}_{\pm0.83}$ | $14.74_{\pm1.73}$ | $14.67_{\pm0.97}$ | $17.13_{\pm0.53}$ |
| | REDDIT | $12.00_{\pm0.49}$ | $9.88_{\pm0.23}$ | $15.63_{\pm0.76}$ | $\mathbf{20.70}_{\pm0.58}$ | $\underline{15.99}_{\pm1.28}$ | $15.65_{\pm1.06}$ | $15.90_{\pm1.86}$ |
| | MOOC | $6.54_{\pm0.79}$ | $5.86_{\pm0.55}$ | $58.95_{\pm1.46}$ | $\underline{59.70}_{\pm2.62}$ | $57.42_{\pm2.80}$ | $\mathbf{62.85}_{\pm2.43}$ | $56.49_{\pm2.72}$ |
| | ENRON | $10.26_{\pm0.18}$ | $13.24_{\pm0.41}$ | $16.80_{\pm6.40}$ | $\underline{42.02}_{\pm7.29}$ | $28.73_{\pm7.35}$ | $\mathbf{48.91}_{\pm7.13}$ | $16.19_{\pm5.91}$ |
| | UCI | $5.13_{\pm1.26}$ | $4.61_{\pm0.86}$ | $20.06_{\pm2.09}$ | $\underline{22.30}_{\pm1.73}$ | $17.58_{\pm3.29}$ | $\mathbf{25.55}_{\pm1.29}$ | $18.63_{\pm2.37}$ |
| | Avg. Rank | 6.20 | 6.80 | 3.20 | **1.80** | 3.60 | 2.40 | 4.00 |
| C | WIKIPEDIA | $10.50_{\pm0.82}$ | $9.71_{\pm0.71}$ | $17.18_{\pm1.04}$ | $\mathbf{19.56}_{\pm1.76}$ | $13.03_{\pm0.98}$ | $15.39_{\pm1.34}$ | $\underline{18.09}_{\pm0.99}$ |
| | REDDIT | $20.15_{\pm0.56}$ | $18.57_{\pm0.23}$ | $50.22_{\pm3.52}$ | $43.89_{\pm9.78}$ | $\mathbf{58.45}_{\pm0.65}$ | $53.61_{\pm1.81}$ | $\underline{54.89}_{\pm2.52}$ |
| | MOOC | $3.50_{\pm0.61}$ | $3.67_{\pm0.22}$ | $\underline{89.48}_{\pm3.04}$ | $69.69_{\pm9.05}$ | $84.67_{\pm5.07}$ | $81.17_{\pm4.73}$ | $\mathbf{91.62}_{\pm1.88}$ |
| | Avg. Rank | 6.33 | 6.67 | 3.00 | 3.67 | 3.00 | 3.67 | **1.67** |
| T-C | WIKIPEDIA | $11.85_{\pm0.38}$ | $11.80_{\pm0.63}$ | $17.74_{\pm1.51}$ | $\mathbf{18.58}_{\pm0.83}$ | $14.04_{\pm1.76}$ | $16.79_{\pm0.84}$ | $\underline{18.16}_{\pm0.78}$ |
| | REDDIT | $11.23_{\pm0.26}$ | $9.78_{\pm0.09}$ | $54.59_{\pm1.98}$ | $46.71_{\pm5.05}$ | $\underline{57.13}_{\pm2.62}$ | $\mathbf{57.47}_{\pm1.60}$ | $48.75_{\pm9.23}$ |
| | MOOC | $6.40_{\pm0.86}$ | $6.20_{\pm0.62}$ | $88.90_{\pm2.75}$ | $84.06_{\pm2.49}$ | $84.33_{\pm6.44}$ | $\underline{89.52}_{\pm2.38}$ | $\mathbf{90.32}_{\pm2.68}$ |
| | Avg. Rank | 6.00 | 7.00 | 3.00 | 3.67 | 3.67 | **2.33** | **2.33** |
| S-C | WIKIPEDIA | $29.97_{\pm0.95}$ | $23.51_{\pm0.94}$ | $25.86_{\pm3.50}$ | $\mathbf{32.15}_{\pm3.68}$ | $11.25_{\pm3.38}$ | $5.45_{\pm0.61}$ | $\underline{31.42}_{\pm2.83}$ |
| | REDDIT | $41.27_{\pm0.43}$ | $24.39_{\pm0.15}$ | $38.53_{\pm3.64}$ | $37.37_{\pm6.90}$ | $\mathbf{52.22}_{\pm0.95}$ | $\underline{41.82}_{\pm3.46}$ | $37.50_{\pm7.67}$ |
| | MOOC | $8.95_{\pm0.23}$ | $8.01_{\pm0.31}$ | $\underline{64.09}_{\pm2.91}$ | $\mathbf{72.80}_{\pm3.55}$ | $61.34_{\pm4.02}$ | $53.69_{\pm5.97}$ | $62.85_{\pm1.97}$ |
| | ENRON | $\mathbf{58.50}_{\pm0.87}$ | $\underline{49.49}_{\pm1.61}$ | $27.52_{\pm8.84}$ | $13.57_{\pm3.81}$ | $6.51_{\pm1.80}$ | $7.77_{\pm0.58}$ | $19.73_{\pm10.93}$ |
| | UCI | $12.65_{\pm0.74}$ | $13.17_{\pm0.79}$ | $6.42_{\pm0.64}$ | $\underline{15.30}_{\pm4.59}$ | $6.35_{\pm0.88}$ | $5.36_{\pm1.07}$ | $\mathbf{17.41}_{\pm3.24}$ |
| | Avg. Rank | 3.40 | 4.80 | 3.60 | **3.00** | 4.80 | 5.40 | **3.00** |
| T-S-C | WIKIPEDIA | $33.50_{\pm0.16}$ | $25.12_{\pm1.52}$ | $37.55_{\pm2.67}$ | $63.17_{\pm3.25}$ | $37.77_{\pm7.05}$ | $\underline{66.33}_{\pm4.35}$ | $\mathbf{76.20}_{\pm2.67}$ |
| | REDDIT | $41.71_{\pm0.17}$ | $24.99_{\pm0.33}$ | $\underline{67.94}_{\pm4.54}$ | $67.34_{\pm4.45}$ | $53.83_{\pm2.43}$ | $63.47_{\pm4.18}$ | $\mathbf{69.49}_{\pm4.71}$ |
| | MOOC | $16.73_{\pm0.29}$ | $10.18_{\pm4.32}$ | $\mathbf{94.26}_{\pm2.84}$ | $90.62_{\pm1.24}$ | $91.80_{\pm2.39}$ | $\underline{93.65}_{\pm2.04}$ | $92.95_{\pm1.74}$ |
| | ENRON | $\mathbf{56.66}_{\pm1.67}$ | $47.39_{\pm3.23}$ | $27.68_{\pm7.80}$ | $49.56_{\pm3.46}$ | $43.19_{\pm4.56}$ | $\underline{54.09}_{\pm12.25}$ | $22.03_{\pm8.71}$ |
| | UCI | $13.91_{\pm2.80}$ | $10.04_{\pm0.96}$ | $27.95_{\pm10.41}$ | $42.88_{\pm7.23}$ | $\underline{45.83}_{\pm3.81}$ | $37.62_{\pm7.88}$ | $\mathbf{53.87}_{\pm2.67}$ |
| | Avg. Rank | 5.00 | 6.40 | 3.80 | 3.40 | 4.00 | 2.80 | **2.60** |

Table 12: AP for models on LANL and DARPA-THEIA with organic anomalies. **Bold** and underlined values mark first- and second-best performing models across rows.

| Datasets | EdgeBank$_\infty$ | EdgeBank$_{tw}$ | TGN | CAWN | TCL | GraphMixer | DyGFormer |
|---|---|---|---|---|---|---|---|
| LANL | 1.01 | 0.82 | 5.77 | 6.13 | 1.11 | 3.85 | **7.08** |
| DARPA-THEIA | 0.01 | 0.01 | 0.07 | 0.02 | 0.01 | 0.04 | **0.08** |
| Avg. Rank | 5.50 | 6.50 | 2.50 | 3.00 | 6.00 | 3.50 | **1.00** |

Table 13: Recall@k for models on LANL and DARPA-THEIA with organic anomalies. **Bold** and underlined values mark first- and second-best performing models across rows.

| Datasets | EdgeBank$_\infty$ | EdgeBank$_{tw}$ | TGN | CAWN | TCL | GraphMixer | DyGFormer |
|---|---|---|---|---|---|---|---|
| LANL | 0.27 | 0.27 | **12.30** | 5.61 | 4.55 | 5.88 | 9.09 |
| DARPA-THEIA | 0.00 | 0.00 | **0.77** | 0.00 | 0.00 | **0.77** | **0.77** |
| Avg. Rank | 5.00 | 5.00 | **1.00** | 4.00 | 4.50 | 2.00 | 1.50 |

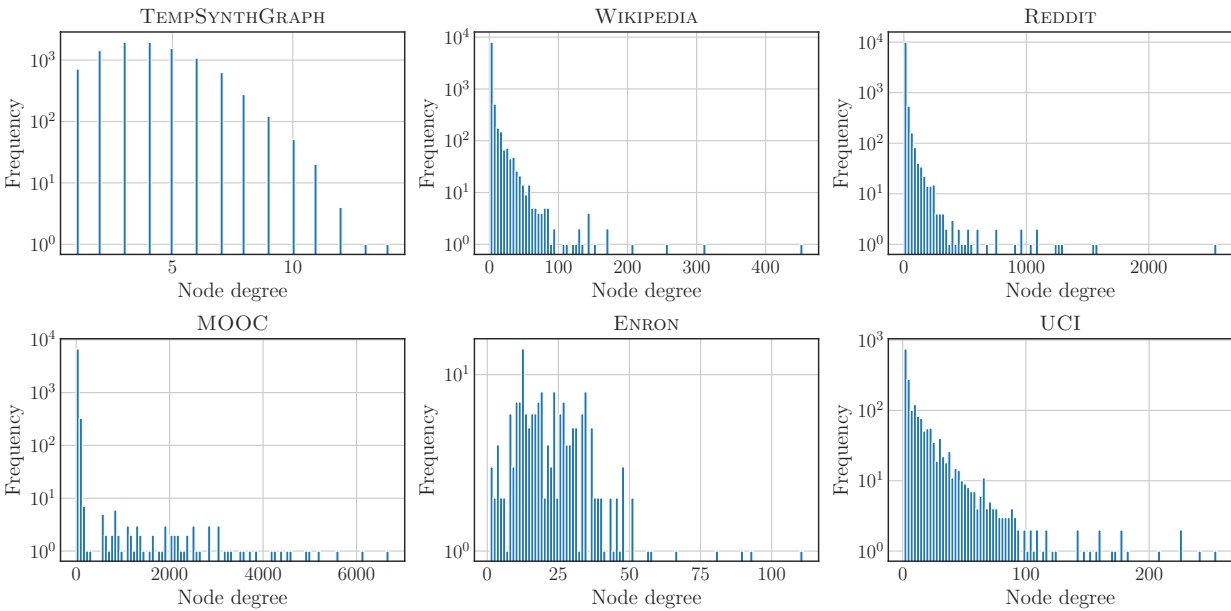

Figure 2: Distributions of node degrees in synthetic and real-world graphs.

# D    Statistical Properties of the Synthetic and Real-World Graphs

We present an analysis of the statistical properties of the synthetic graphs introduced in §3.3 and the real-world graphs listed in App. A.1. In particular, we analyze properties related to the structure, context, and temporal dynamics of the graphs, highlighting consistencies and irregularities in and across these three dimensions.

**Structure**. To analyze and compare the graphs for their structural properties, we compute the distributions of node degrees and the joint degree matrices, defined as the relative number of edges between vertices of degree $i$ and degree $j$ for every pair $(i, j)$, of the static versions of the graphs (i.e., the graphs without timestamps). These statistics are displayed in Figure 2 and Figure 3, respectively. Notably, both statistics reveal Gaussian distributions of and across node degrees in the synthetic graph, confirming the desired structural consistency in this controlled setting. Conversely, distributions of and across node degrees are less structured within and inconsistent across the real-world datasets. Notably, all real-world datasets have a long-tailed distribution of node degrees, with top degrees ranging from 111 (ENRON) to 6695 (MOOC). Moreover, node degrees in WIKIPEDIA, REDDIT, and UCI follow exponential distributions, whereas node degrees in the ENRON dataset follow a noisy Gaussian distribution. MOOC is globally weakly interconnected except for various outlier nodes with degrees between 2 and 6695. Moreover, the similarity between the degrees of neighboring nodes constituting the joint degree matrices depicted suggests the datasets' structural homogeneity and consistency, which Figure 3 reveals to be strong in TEMPSYNTHGRAPH, weaker yet existing in ENRON and UCI, and at most very weak in WIKIPEDIA, REDDIT, and MOOC.

**Context**. We investigate the contextual consistencies of graphs by analyzing the distributions of standard deviations $\sigma$ of edge messages across timestamps. Higher standard deviations indicate edges with inconsistent edge messages over time. In this view, Figure 4 shows that edge messages in real-world graphs exhibit substantially lower contextual consistency than the synthetic graph. Specifically, TEMPSYNTHGRAPH features standard deviations concentrated around 0.05, whereas WIKIPEDIA, REDDIT, and MOOC feature exponential distributions of standard deviations with tails ending around 9, 5, and 20, respectively. Note that ENRON and UCI have standard deviations of zero due to the absence of edge messages in these datasets.

**Time**. We analyze the temporal patterns of the graphs by examining the distributions of standard deviations of inter-event times for edges that reoccur multiple times. Wider distributions indicate graphs with less

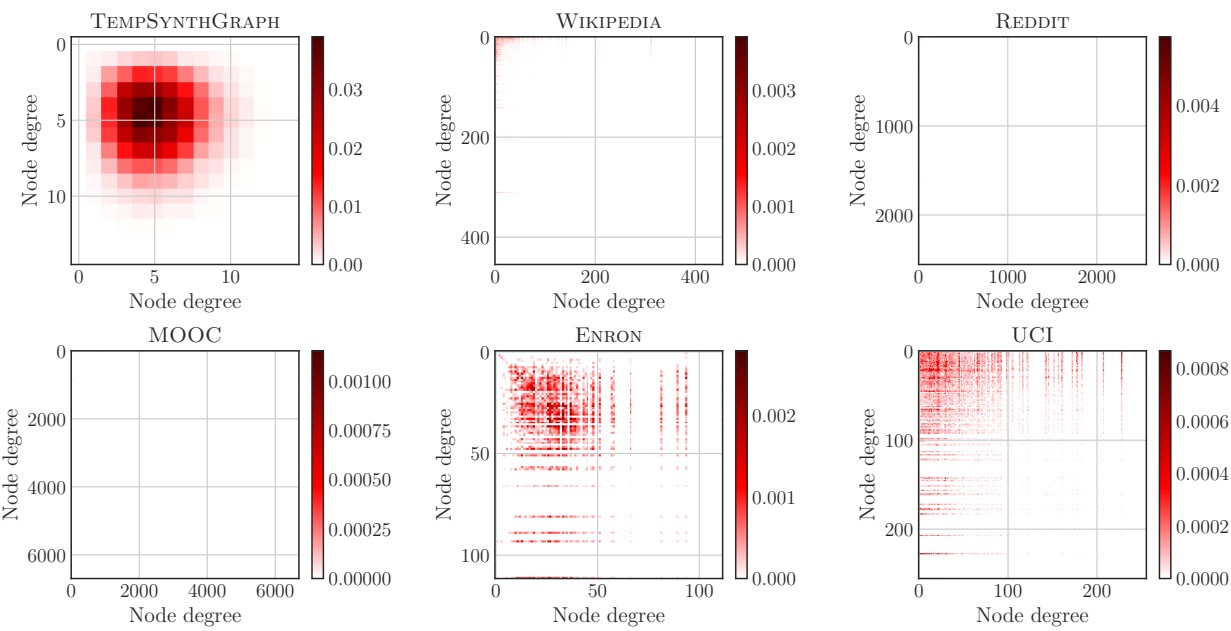

Figure 3: Joint degree matrices of synthetic and real-world graphs.

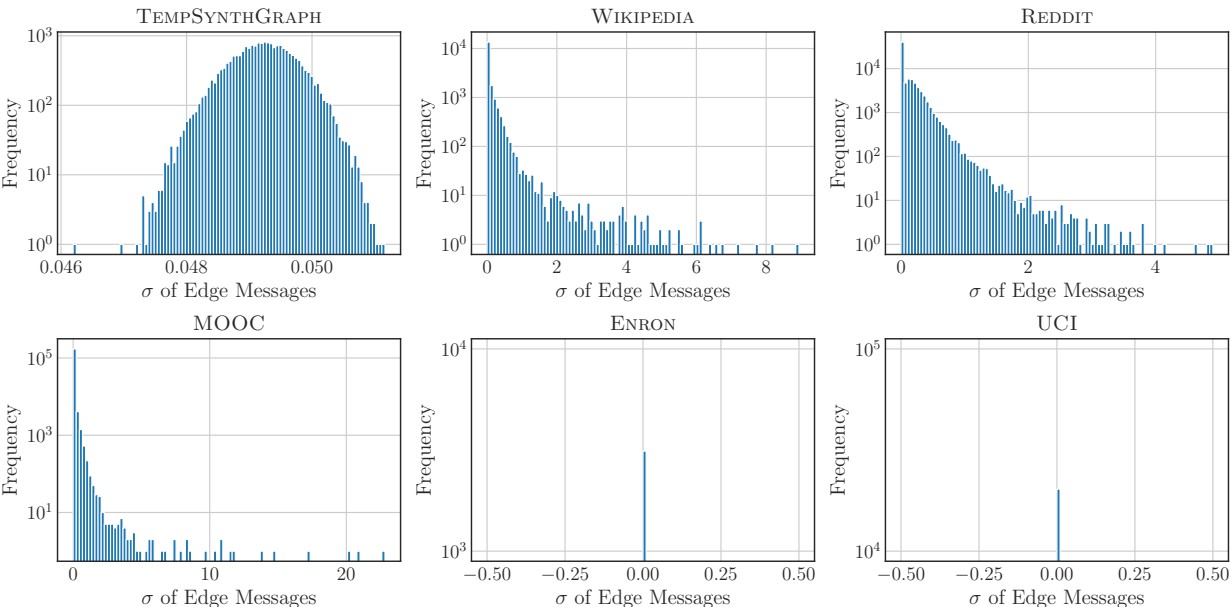

Figure 4: Distributions of the standard deviations of edge messages across timestamps for edges in synthetic and real-world graphs.

consistent intervals between re-occurrences of the edges. As shown in Figure 5, inter-event times are notably less consistent in real-world graphs compared to the synthetic one. Specifically, TEMPSYNTHGRAPH exhibits standard deviations below $10^4$, whereas the standard deviations for WIKIPEDIA, REDDIT, MOOC, ENRON, and UCI span multiple orders of magnitude, reaching up to $10^7$.

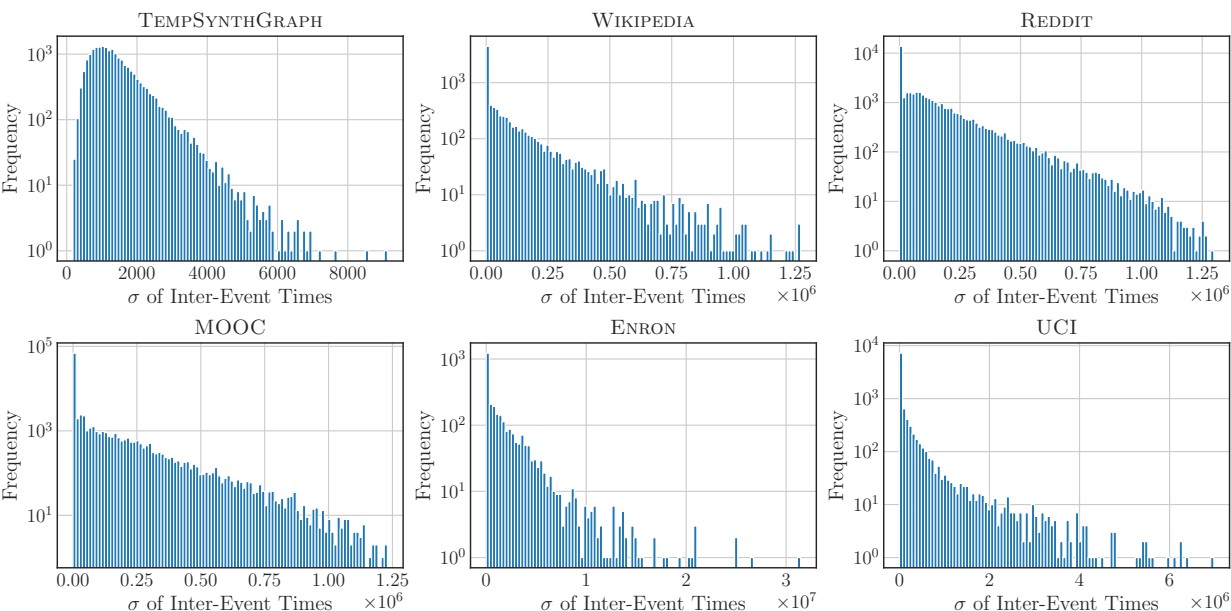

Figure 5: Distributions of the standard deviations of inter-event times for edges in synthetic and real-world graphs.

