# OpenReview forum: "Learning-Based Link Anomaly Detection in Continuous-Time Dynamic Graphs"
_TMLR — Accepted by TMLR_

### Review · Reviewer_KNpY · 2024-06-21

**Summary Of Contributions:**

The authors proposed a benchmark for anomaly detection in continuous-time dynamic graphs (CTDG), including a new taxonomy and algorithms for generating synthetic CTDGs and injecting corresponding anomalies. In the experiments, they adapted and compared seven existing CTDG learning methods using this benchmark, and gave analysis of the experimental results.

**Audience:**

Yes

**Broader Impact Concerns:**

There is no ethical implications of this work.

**Claims And Evidence:**

Yes

**Requested Changes:**

1. The authors need to enhance their motivation of proposing the new taxonomy. I understand that existing taxonomies are not comprehensive - as almost all taxonomies are - I am looking for explanations that clarify what inspired the authors to design this new taxonomy in such a specific way. For examples, *"previous taxonomies cannot adequately address the studied anomaly detection tasks on CTDGs, and there are emerging real-world graph applications that necessitate detecting edge-level anomalies with the proposed fine-grained detail."* Meanwhile, the authors need to emphasize the importance of the studied problem (edge-level anomaly detection), for instance, *what is its application?*
2. The authors need to clarify their motivation for introducing a synthetic CTDG generation algorithm. I understand that the algorithm ensures consistencies across multiple dimensions, but it is unclear why such perfect consistency is deemed essential for the task. Expecially, real-world CTDG datasets do not exhibit such perfect consistency.
3. In **Definition 1** (Continuous-time dynamic graph), is the "non-decreasing" hypothesis a conventional setting for CTDG? I believe some works account for scenarios where edges are removed over time [1], where the time attribute in interactions is represented as time spans rather than timestamps. If it is not conventional, the authors should highlight that supposing "non-decreasing" is specific to this work.
4. In Section 5.4, where the authors concluded **"real graphs may not exhibit strong consistencies"**, the authors are suggested to present statistical differences of synthetic and real graphs to support this conclusion. For example, *their distributions of node degrees and temporal attributes*.

[1] Nguyen et al. Continuous-Time Dynamic Network Embeddings, WWW'18, https://dl.acm.org/doi/10.1145/3184558.3191526

**Strengths And Weaknesses:**

**Strengths**: The authors conducted a thorough review of the related work, analyzing their limitations. The proposed benchmark, including a fine-grained taxonomy and dataset generation algorithms, is clearly described.

**Weaknesses** (see *Requested Changes* for details):
1. The necessity for proposing the new taxonomy is not well highlighted or convincing. While the authors discuss the limitations of existing taxonomies for anomalies in dynamic graphs, I would appreciate examples of real applications on CTDGs that necessitate the new taxonomy introduced in this paper.
2. The motivation behind introducing algorithms for generating synthetic CTDG and injecting anomalies are not well conveyed.
3. The significance of the studied problem (edge-level anomaly detection) is not well conveyed.
4. Some statements and conclusions require more supporting evidence.

---

> ### Author Response · Authors · 2024-07-24
> **Addressing of all questions and change requests**
>
> We thank the reviewer for their detailed feedback. Here we respond to their questions one by one:
>
> **Response to Q1**: Graph properties, including the dimensions along and across which consistency (normality) and anomalies are expected and observed, vary across applications. Our motivation for this taxonomy for typed anomalies is three-fold:
> 1. Incompleteness of existing taxonomies, e.g., [8], [9] - expanded on in the Introduction.
> 2. Intuitiveness of the proposed taxonomy, which is simple yet rich enough to model anomalies in any sensible combination of structure, context, and time.
> 3. Real-world anomalies categorically describable using this taxonomy, such as:
>
>    3.1. Fault detection in manufacturing (inconsistencies in time).
>
>    3.2. Social networks, where hacked accounts post untypical content (contextual inconsistency).
>
>    3.3. Hacked email accounts sending unusual emails (content) to unusual recipients (structure), possibly outside of working hours (time).
>
>    3.4. Fraud detection in financial networks, where large amounts of money (context) is sent to unexpected recipients (structure).
>
>    3.5. Data breaches in ICT networks, where large data (context) transfers using unusual protocols (also context) to unexpected recipients (structure) at low activity times (time).
>
> Finally, our taxonomy and synthetic generation processes allow practitioners to analyze, compare, and choose suitable methods for detecting the typed anomalies they expect to encounter in their applications.
>
> **Response to Q2**: Our CTDG generation process serves three objectives:
> 1) Creating an ideal environment where graph consistencies and typed anomalies can be controlled and analyzed in isolation and co-occurrence;
> 2) Assessing the capabilities of link prediction methods to capture consistencies and detect the five anomaly types defined by our taxonomy;
> 3) Validating our taxonomy and procedures for synthesizing and injecting edge-level anomalies.
>
> This process enabled scientific experimentation and drawing of conclusions within our scope, facilitating future research on CTDGs beyond anomaly detection, including other prediction tasks. Appendix D displays the variation in consistencies measured for real-world graphs compared to synthetic CTDGs, confirming that non-synthetic settings do not allow for analyses of typed consistencies and anomalies in isolation or co-occurrence.
>
> **Response to Q3**: We acknowledge that CTDG definitions vary across the literature. Some works [1, 2] treat edge disappearance as explicit events. However, most recent research [3, 4, 5, 6, 7] models edges as instantaneous events, allowing for the modeling of edge lifespans by modelling event types such as "edge start/appearance" and "edge end/disappearance" as a categorical feature within the edge's message. Given the non-increasing nature of time, the assumption of events occurring in chronological order causes no loss of generality. Existing link prediction methods and CTDG datasets used in our work also consider links as instantaneous events. Thus, we prefer to stick to following the common practice of arranging edges in non-decreasing order based on their timestamps.
>
> **Response to Q4**: We appreciate the suggestion to support our claim that real graphs may not exhibit strong consistencies with statistical evidence. Following this suggestion, we have added comprehensive statistical analyses of synthetic and real-world graphs in a new section (Appendix D). Specifically, we included plots and descriptions of the following statistics, displaying discrepancies in consistency levels in structure, context, and time for synthetic and real-world graphs:
>
> - Structure: Node degree distribution and joint degree matrices.
> - Context: Standard deviations of edge messages across timestamps.
> - Time: Standard deviations of inter-event times for edges.
>
> We believe that these requested changes further clarify our claims and support the discussions about structural, contextual, and temporal consistencies and the corresponding types of anomalies in synthetic and real-world graphs.
>
> **References**
>
> - [1] Nguyen et al. "Continuous-time dynamic network embeddings" WWW'18.
> - [2] Rossi et al. "Temporal graph networks for deep learning on dynamic graphs" arXiv'20.
> - [3] Wang et al. "Inductive Representation Learning in Temporal Networks via Causal Anonymous Walks" ICLR'21.
> - [4] Kumar et al. "Predicting dynamic embedding trajectory in temporal interaction networks" SIGKDD'19.
> - [5] Cong et al. "Do We Really Need Complicated Model Architectures For Temporal Networks?" ICLR'22.
> - [6] Yu et al. "Towards better dynamic graph learning: New architecture and unified library" NeurIPS'23.
> - [7] Poursafaei et al. "Towards better evaluation for dynamic link prediction" NeurIPS'22.
> - [8] Liu et al. "Benchmarking unsupervised outlier node detection on static attributed graphs" NeurIPS'22.
> - [9] Ma et al. "A comprehensive survey on graph anomaly detection with deep learning" IEEE TKDE'21.

---

### Review · Reviewer_emDe · 2024-07-01

**Summary Of Contributions:**

The authors propose a framework for anomaly detection on edges of dynamic graphs.  In addition to tackling a very interesting and challenging research problem, the authors main contributions are:
* They propose a framework that leverages structural, temporal, and contextual features for dectecting edge-level anomalies on graphs
* On the back of such framework, the authors propose methods for injecting anomalies into graphs as well as generating continuous-time dynamic graphs
* They extend link prediction learning methods by conditioning edge existence on contextual edge attributes
* They perform extensive experiments with synthetic and real-world dataset that show strong evidence for validation of the proposed methods

In summary, tasks revolving dynamic graphs estimation (i.e., dynamic link prediction, node fault dectection, graph forecasting, time-varying graph clustering) are challenging and underexplored topics; in this manuscript, the authors made a meaningful step towards advancing the practical knowledge on the field.

**Audience:**

Yes

**Broader Impact Concerns:**

No concerns on ethical implications of the work.

**Claims And Evidence:**

Yes

**Requested Changes:**

As I mentioned previously, it would be relevant to have a paragraph discussing potential alternatives for incorporating the additional edge contextual information into the graph training process.

**Strengths And Weaknesses:**

Strengths
=======

* Authors presented a novel taxonomy for edge anomaly types that enhances current understanding on synthetic graph generation and anomaly dectection in continuous time dynamic graphs.

* Extensive experiments on real and synthetic datasets help to elucidate the advantages of the proposed methodology.


Weaknesses
=========

* Apart from the last paragraph on section 4.2, little has been discussed on possible other potential alternatives for incorporating contextual information into the training process.  Are there any other options? If so, how does it compare the one used in paper?

---

> ### Author Response · Authors · 2024-07-24
> **Alternatives for incorporating the additional edge contextual information into the graph training process**
>
> We thank the reviewer for their detailed feedback and acknowledgment of the novelty of our proposed taxonomy as well as the contributions of our work. Here we address the question about alternatives for incorporating the additional edge contextual information into the graph training process:
>
> The method presented in this paper concatenates node embeddings with a linear projection of the edge attributes before feeding the concatenation to the prediction head. The approach draws inspiration from existing temporal graph learning methods, where concatenation between features (or embeddings) is the de facto preferred way to merge information from various sources. Another straightforward approach would be summation of node embeddings and projected contextual attributes, although this is generally less effective than concatenation. Works such as [1,2,3,4] are well-known examples of using concatenation to merge various sources of information in continuous time dynamic graphs.
>
> Less straightforward approaches to incorporating contextual information could include: (1) domain-specific solutions (such as gates or Hadamard products [5, 6]) or (2) integrating the context into the message-passing process of each model. For the sake of generality, our paper does not investigate these alternatives, as they are domain-specific and might require different design choices for each dataset or use case.
>
> We updated the manuscript with some clarifications regarding this aspect.
>
> - [1] Rossi, Emanuele, et al. "Temporal graph networks for deep learning on dynamic graphs." arXiv preprint arXiv:2006.10637 (2020).
> - [2] Cong, Weilin, et al. "Do We Really Need Complicated Model Architectures For Temporal Networks?." The Eleventh International Conference on Learning Representations.
> - [3] Yu, Le, et al. "Towards better dynamic graph learning: New architecture and unified library." Advances in Neural Information Processing Systems 36 (2023): 67686-67700.
> - [4] Xu, Da, et al. "Inductive representation learning on temporal graphs." arXiv preprint arXiv:2002.07962 (2020).
> - [5] Beutel, Alex, et al. "Latent cross: Making use of context in recurrent recommender systems." Proceedings of the eleventh ACM international conference on web search and data mining. 2018.
> - [6] Kumar, Srijan, Xikun Zhang, and Jure Leskovec. "Predicting dynamic embedding trajectory in temporal interaction networks." Proceedings of the 25th ACM SIGKDD international conference on knowledge discovery & data mining. 2019.

---

### Review · Reviewer_fVbf · 2024-07-11

**Summary Of Contributions:**

This paper addresses the lack of existing methods for anomaly detection in continuous-time dynamic graphs, particularly those dealing with various types of anomalies (contextual, structural, and temporal). The authors propose several key contributions: they introduce a method to generate and inject typed anomalies into graphs. By doing so, they create realistic scenarios for evaluating anomaly detection algorithms.
Also, the paper extends the optimization problem of link prediction to that conditioning link existence on contextual edge attributes.
Then they provide a comprehensive experiment that benchmarks many SOTA models on both synthetic and real-world datasets.

**Audience:**

Yes

**Claims And Evidence:**

Yes

**Requested Changes:**

None

**Strengths And Weaknesses:**

**Strengths:** The paper is well-written and easy to understand. It provides clear definitions of different types of anomalies with straightforward examples. Furthermore, it clearly distinguishes between link prediction and link anomaly detection, emphasizing the importance of considering different types of anomaly detection. The paper also introduces effective negative edge sampling methods to address this challenge, enabling state-of-the-art (SOTA) models to perform well on these tasks.

**Weakness:** The paper does not provide sufficient empirical comparisons between considering different types of anomalies and not doing so. While I understand that this comparison might be challenging, I believe it would significantly strengthen the proposed work.

---

> ### Author Response · Authors · 2024-07-24
> **Empirical analyses on typed vs. untyped edge anomaly detection**
>
> We thank the reviewer for their comments, acknowledgment of both the contributions of our work and the clarity of the presentation, and their suggestion to add further empirical analysis in support of the comparison between typed and untyped edge anomaly detection. In response to the latter point, we would like to refer to two experiments presented in Section 5.2 that outline the advantages of considering typed anomalies at training time. These experiments, reported in Table 2, show significant benefits in AUC when using the adaptations proposed by our method in comparison to an anomaly-unaware, i.e., unitype approach. Additionally, we agree with the reviewer that extending the manuscript with more comprehensive empirical comparisons might complement the existing content, particularly that presented in Section 5. Nonetheless, we consider the quantity and quality of the statistical analyses and experiments, including those we newly added to the manuscript in response to suggestions made by reviewer KNpY (see Appendix D), adequate to support the theoretical claims while retaining a desirable level of conciseness.
>
> To further address the reviewer’s point on typed vs. untyped link anomalies, we would like to refer to our response to Q1 of reviewer KNpY, where we elucidate the importance of the particular types of anomalies defined by our taxonomy and list applications in which anomalies clearly fall into these categories.
>
> Finally, our work on typed edges, including the edge anomaly injection procedures, enables practitioners to analyze and compare methods for their capabilities in detecting the types of anomalous links they expect to encounter in their application, and thus supports them in selecting the best models for the domain-specific data at hand - which has constituted a gap in the current literature.

---

### Decision · Action_Editor_vNDK · 2024-08-28

**Recommendation:** Accept as is

**Comment:**

The paper was reviewed by three expert reviewers. The reviewers agreed that the paper provides clear definitions of the different types of anomalies and a thorough experimental evaluation on both synthetic and real-world datasets. However, the reviewers also raised concerns mainly about the motivation of the work (do we actually need a new taxonomy?), the intuition behind the proposed taxonomy (why is this the right taxonomy and not others?) and the considered setting (a continuous-time dynamic graph is defined as a sequence of non-decreasing chronological interactions).  The authors comprehensively responded to the reviewers' comments, and all reviewers are now positive about the paper. I thus think that the paper is ready for publication. Since regular submissions consist of no more than 12 pages of main content, I request the authors to make sure that the conclusion fits within the 12-page limit.

**Audience:**

Several important applications (e.g., sensor fault detection, fraud identification) could benefit from new developments in the field of anomaly detection in continuous-time dynamic graphs. Thus, the paper's findings will be of interest to some individuals in TMLR's audience.

**Claims And Evidence:**

This paper deals with the problem of anomaly detection in continuous-time dynamic graphs. First, a taxonomy for edge-level anomalies is introduced along with a method for generating and injecting typed anomalies into graphs. Then, some tricks are proposed such that link prediction algorithms can be used for anomaly detection. Experimental results demonstrate that these tricks indeed lead to performance gains in the task of anomaly detection. The empirical results also demonstrate that there is no single learning algorithm that provides a superior performance under all types of anomalies.